palaeontology, evolution, ecology

biodiversity, Tetrapoda, Phanerozoic, terrestrial, diversification, palaeontology

**Authors for correspondence:**
Roger A. Close
e-mail: r.a.close@bham.ac.uk
Richard J. Butler
e-mail: r.butler.1@bham.ac.uk

# The apparent exponential radiation of Phanerozoic land vertebrates is an artefact of spatial sampling biases

Roger A. Close[1], Roger B. J. Benson[2], John Alroy[3], Matthew T. Carrano[4], Terri J. Cleary[1], Emma M. Dunne[1], Philip D. Mannion[5], Mark D. Uhen[6] and Richard J. Butler[1]

[1]School of Geography, Earth and Environmental Sciences, University of Birmingham, Edgbaston, Birmingham B15 2TT, UK
[2]Department of Earth Sciences, University of Oxford, Oxford OX1 3AN, UK
[3]Department of Biological Sciences, Macquarie University, NSW 2109, Australia
[4]Department of Paleobiology, National Museum of Natural History, Smithsonian Institution, Washington, DC 20013, USA
[5]Department of Earth Sciences, University College London, London WC1E 6BT, UK
[6]Department of Atmospheric, Oceanic, and Earth Sciences, George Mason University, Fairfax, VA 22030, USA

RAC, 0000-0003-3302-9902; RBJB, 0000-0001-8244-6177; JA, 0000-0002-9882-2111;
MTC, 0000-0003-2129-1612; TJC, 0000-0003-0424-8073; EMD, 0000-0002-4989-5904;
PDM, 0000-0002-9361-6941; MDU, 0000-0002-2689-0801; RJB, 0000-0003-2136-7541

There is no consensus about how terrestrial biodiversity was assembled through deep time, and in particular whether it has risen exponentially over the Phanerozoic. Using a database of 60 859 fossil occurrences, we show that the spatial extent of the worldwide terrestrial tetrapod fossil record itself expands exponentially through the Phanerozoic. Changes in spatial sampling explain up to 67% of the change in known fossil species counts, and these changes are decoupled from variation in habitable land area that existed through time. Spatial sampling therefore represents a real and profound sampling bias that cannot be explained as redundancy. To address this bias, we estimate terrestrial tetrapod diversity for palaeogeographical regions of approximately equal size. We find that regional-scale diversity was constrained over timespans of tens to hundreds of millions of years, and similar patterns are recovered for major subgroups, such as dinosaurs, mammals and squamates. Although the Cretaceous/Palaeogene mass extinction catalysed an abrupt two- to three-fold increase in regional diversity 66 million years ago, no further increases occurred, and recent levels of regional diversity do not exceed those of the Palaeogene. These results parallel those recovered in analyses of local community-level richness. Taken together, our findings strongly contradict past studies that suggested unbounded diversity increases at local and regional scales over the last 100 million years.

## 1. Introduction

Life on land today is spectacularly diverse, accounting for 75–85% of all species [1,2]. Understanding how terrestrial diversity was assembled through deep time is crucial for settling fundamental debates about the diversification process, such as whether it is constrained by ecological limits [3,4]. However, there is no consensus about the long-term trajectory of terrestrial diversity—in particular, whether or not exponential increases occured through the Phanerozoic, leading to diversity being higher today at local, regional and global scales than at any point in the geological past [3,5–11].

Tetrapods today comprise greater than 30 000 extant species and include many of the most iconic and intensely studied groups of animals. Curves of global Phanerozoic tetrapod palaeodiversity have been widely used as exemplars of

terrestrial diversification [3,7,9]. In particular, they have been used to argue for an 'expansionist' model of diversification, characterized by unconstrained and apparently exponential increases in diversity at a variety of spatial scales, perhaps even driving a tenfold rise in species richness during the last 100 million years (Myr) [7,8]. Within this paradigm, mass extinctions act only as short-term setbacks within a trend of ever-increasing diversity. This expansionist interpretation of terrestrial diversity through deep time has been cited as evidence that contradicts a role for ecological limits in constraining diversification [3] and to propose fundamentally different diversification processes in the marine and terrestrial realms [8].

However, the only diversity curves spanning the entire Phanerozoic evolutionary history of tetrapods are based on first and last appearance data for families, drawn from compilations that are now decades old [5,9]. Families are defined inconsistently [12,13] and may not reflect the patterns of diversity at the species level. Moreover, these curves do not account for pervasive and long-established spatial and temporal sampling biases [14–16], because they predate the widespread use of sampling standardization methods.

Most problematically of all, 'global' palaeodiversity curves based on the worldwide fossil record are not truly global, because the spatial extent of the fossil record varies substantially among intervals of geological time [10,11]. In reality, the 'global' fossil record comprises a heterogeneous set of regional assemblages, with palaeogeographical regions that vary markedly in number, identity and extent (both within and between continental regions) through the intervals of geological time. Critically, the palaeogeographical spread (=spatial extent) of the terrestrial fossil record itself grows exponentially through the Phanerozoic (figure 1b and figure 2; see also the electronic supplementary material, figures S1 and S2) and is decoupled from the actual terrestrial area that existed through time (see Results). Such changes in the geographical extent of the sampled fossil record will substantially bias patterns of diversity through time, even when using sampling-standardized richness estimators [20].

Patterns inconsistent with expansionism are recovered by analyses applying rigorous sampling standardization to estimate regional diversity of more restricted groups of tetrapods [6,21–23] or over shorter intervals of time (the Mesozoic–early Palaeogene; [10,11]). Analyses of Phanerozoic tetrapod diversity at the local community scale [24] also contradict the expansionist model of diversification. However, it remains unclear how terrestrial tetrapod diversity at regional spatial scales changed through the entirety of the Phanerozoic, especially from the Palaeogene to the present, when the most substantial increases in face-value 'global' curves are observed.

Here, we present, to our knowledge, the first regional-scale diversity patterns for terrestrial tetrapods that cover their entire Phanerozoic evolutionary history, while adequately correcting for key biases. In doing so, we interpret the structure of the fossil record as an array of well-sampled palaeogeographical regions that contain useful information about regional palaeodiversity, but which are only indirectly informative about true global palaeodiversity. To achieve this, we extend and substantially improve our recently developed approach for addressing large-scale spatial sampling biases [11]. We conduct our analyses at the species level and compare our results to different models of the diversification process. Our results demonstrate that diversity curves based on face-value counts of taxa from the 'global' fossil record primarily reflect major increases in

the geographical spread of fossil localities towards the present day. After controlling for these biases, we find no evidence for expansionist diversification in regional assemblages. The similarity of this regional pattern to patterns of local richness [24] suggests that beta diversity is unlikely to have changed substantially over the Phanerozoic, although further work is needed to confirm this. These results imply that the global diversity present in terrestrial ecosystems today may be similar to that of the early Cenozoic.

## 2. Material and methods

### (a) Overview of analytical procedure

We estimated diversity and other variables for palaeogeographical regions with approximately equal sizes. To achieve this, our analysis implemented the following steps (each described in more detail below).

(i) We downloaded occurrence data for Phanerozoic non-flying tetrapods and key subgroups from the Paleobiology Database (figure 3a; electronic supplementary material, figure S1), removed unsuitable records and binned the remaining records within equal-length time intervals.

(ii) We used a spatial subsampling algorithm (described below) to identify all nested subsets of adjacent fossil localities (=subsampled palaeogeographical regions) for each time interval, using the set of palaeocoordinates for all collections yielding non-flying terrestrial tetrapods (figure 3c).

(iii) We computed variables of interest (diversity, spatial metrics, etc.) for each subsampled palaeogeographical region.

(iv) We standardized the spatial extent of sampling in the fossil record by identifying subsampled palaeogeographical regions that simultaneously met a set of criteria related to spatial extent (summed minimum-spanning tree (MST) length) and other spatial and sampling-related metrics (see below). This was performed at several distinct spatial scales.

(v) We identified the clusters of overlapping palaeogeographical regions (figure 3d; see below). This is necessary because palaeogeographic regions identified via the exhaustive search algorithm implemented in step (ii) may share many of the same underlying fossil localities.

(vi) All variables computed for palaeogeographical regions were summarized for each spatial cluster by computing medians and interquartile ranges.

### (b) Dataset

We downloaded fossil occurrence data for Phanerozoic Tetrapodomorpha from the Paleobiology Database [25] on 27 February 2019. We also downloaded occurrences for key tetrapod subgroups (Dinosauromorpha, Probainognathia, Squamata, Pseudosuchia, Testudinata and Lissamphibia) and used the 'occurrence_no' fields from these downloads to filter records from the main occurrence dataset. All occurrence datasets were downloaded using the Paleobiology Database API [26], using function calls executed within the analysis R scripts (URLs used to perform these data downloads, together with all analysis scripts, are available on Dryad [27]).

We removed unsuitable records from the occurrence dataset largely following the procedures outlined in Close et al. [11]. Contrary to that study, however, we did not exclude collections from deposits that were unlithified or partially lithified and sieved (this is because lithification biases more severely affect the face-value estimates of local richness analysed by Close et al. [11]). The patterns we document here are therefore

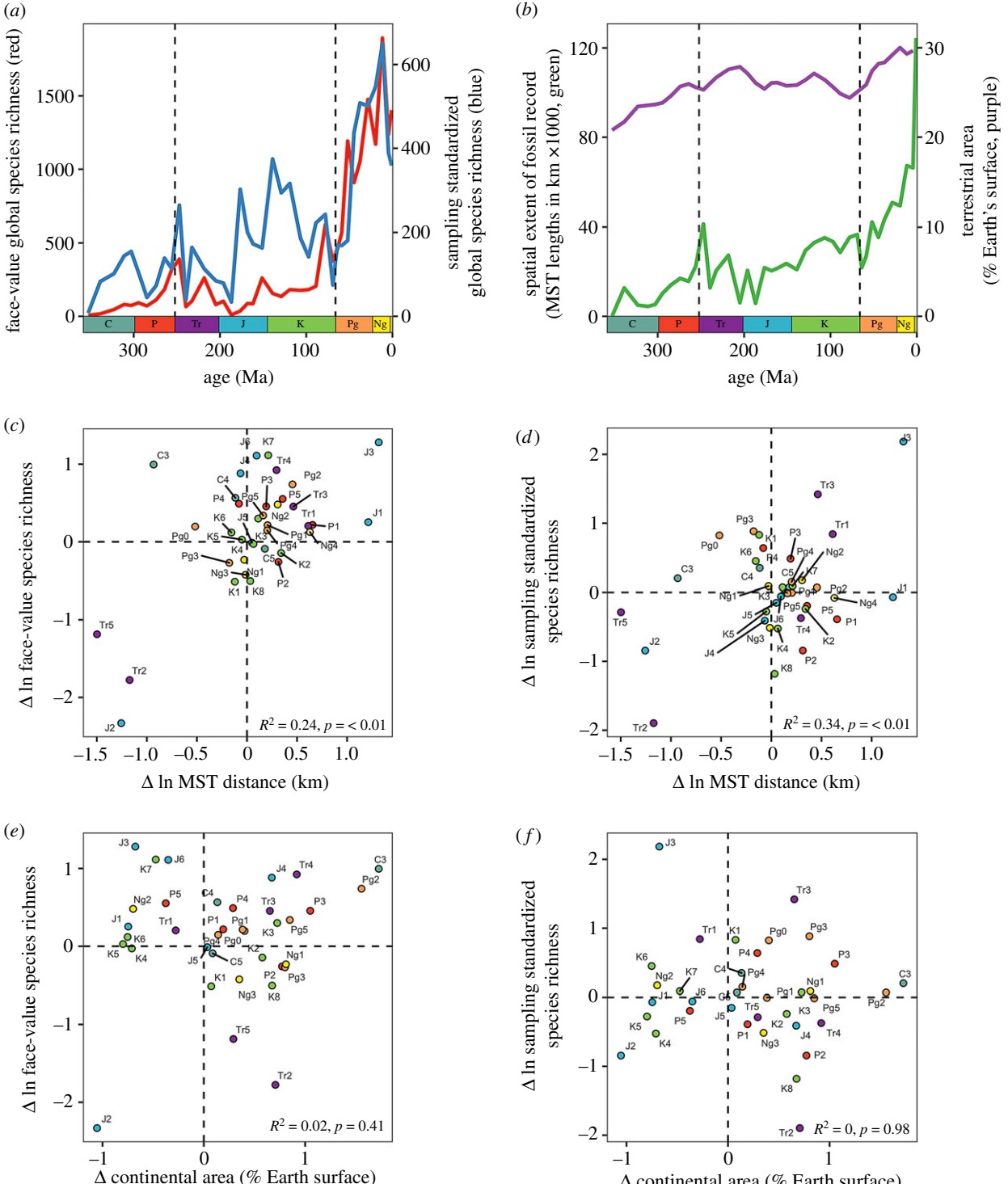

**Figure 1.** Spatial bias and the global fossil record of Phanerozoic terrestrial tetrapods. (*a*) Face-value (red) and sampling-standardized (shareholder quorum subsampling (SQS) [17,18] using quorum=0.6; blue) 'global' species richness of Phanerozoic terrestrial tetrapods. (*b*) Spatial sampling (occupied equal-area grid cells with 500 km spacings, green) and habitable area (terrestrial area as a percentage of Earth's surface [19], purple). Counts of occupied grid cells increase steeply through the Cenozoic and accelerate towards the present. (*c,d*) Relationships between changes in (*c*) face-value and (*d*) sampling-standardized species richness (using SQS, quorum = 0.6) and changes in counts of occupied grid cells per equal-length bin (all variables log-transformed). (*e,f*) Relationships between (*e*) changes in face-value and (*f*) sampling-standardized species richness (using SQS, quorum = 0.6) and changes in continental area through time. Datapoints for C1 and C2 removed as outliers.

conservative with respect to lithification biases, which manifest primarily from the Late Cretaceous onwards and become more profound towards the present. Flying tetrapods (Aves, Pterosauromorpha and Chiroptera) were excluded because their fossil record is inadequate in most intervals and regions, and Lagerstätten-dominated. After cleaning, the dataset comprised 17 323 collections (broadly equivalent to fossil localities; see discussion in [24] for more detail), yielding 60 859 occurrences of 14 023 non-flying, non-marine tetrapod species.

Following previous studies (e.g. [11]), we used composite time bins of approximately equal length (approx. 10 Myr; electronic supplementary material, table S1). Occurrences were assigned to a bin if that bin contained over 50% of the geological time range associated with that occurrence (defined by the early and late bounds recorded by the 'min_ma' and 'max_ma' fields in the Paleobiology Database, in Ma). A total of 4056 occurrences were dropped because they did not meet these binning criteria (72 413 before and 68 357 after).

Proc. R. Soc. B 287: 20200372

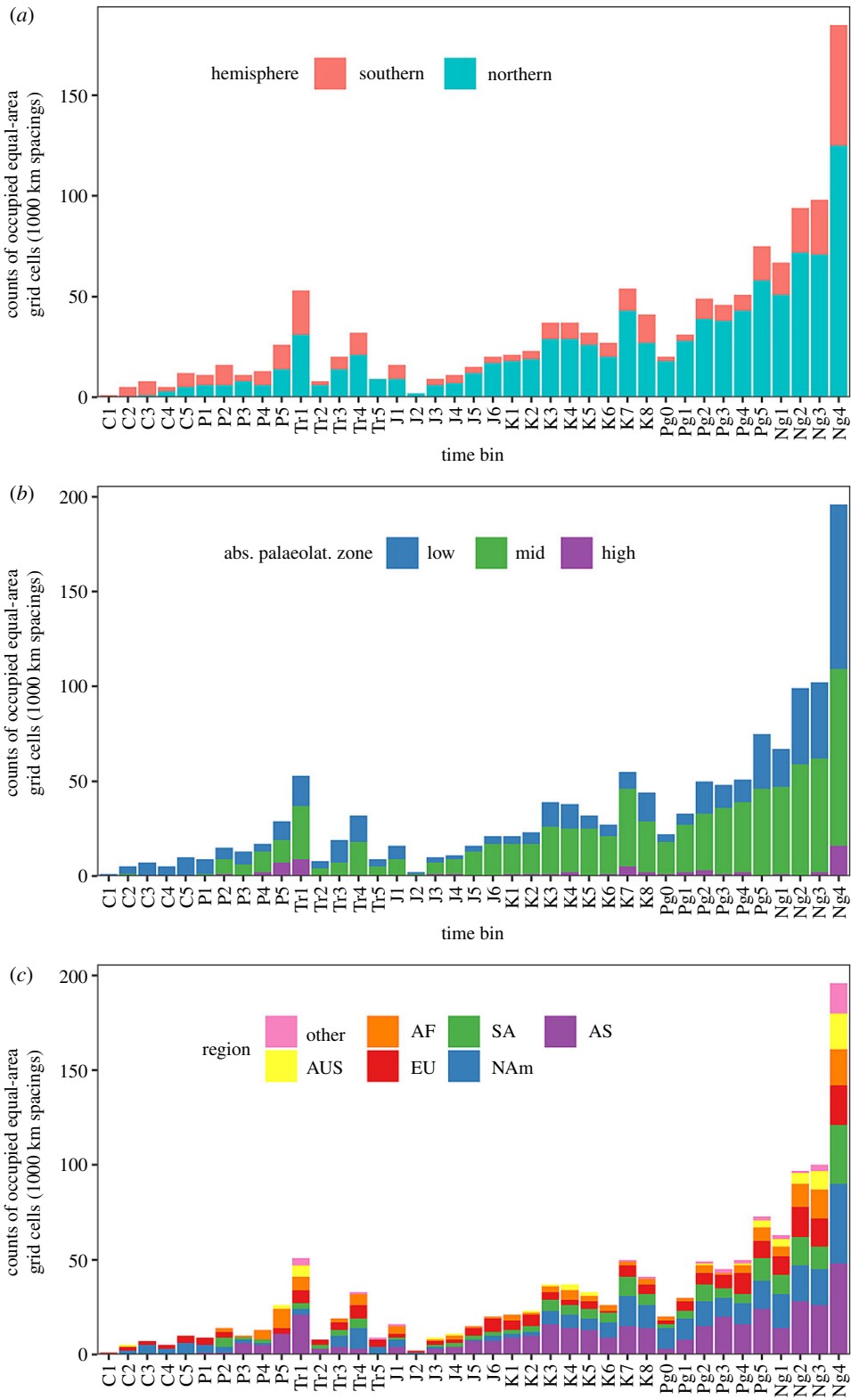

**Figure 2.** Spatial sampling in the Phanerozoic record of terrestrial tetrapods. Per-bin counts of equal-area grid cells with 1000 km spacings, broken down by (*a*) hemisphere, (*b*) absolute palaeolatitude zone (low = 0–30°, mid = 30–60°, high = 60–90°), and (*c*) continental region. Spatial sampling rises steeply through the Phanerozoic and is especially limited outside of North America, Europe and Asia, in the southern hemisphere, and at low and high palaeolatitudes. NAm, North America; EU, Europe; SA, South Africa; AF, Africa; AS, Asia; AUS, Australasia.

## (c) Identifying subsampled palaeogeographical regions

To control for the pervasive spatial sampling biases affecting the terrestrial fossil record, we estimated diversity and other key variables for approximately equally sized palaeogeographical regions, which we defined by drawing spatial subsamples of adjacent fossil localities (on a per-interval basis). To define these palaeogeographical regions, we used a spatial subsampling algorithm that identifies all nested sets of adjacent spatial points [28]. Spatial points were defined by binning the palaeocoordinates for all collections in our cleaned occurrence dataset into equal-size hexagonal/pentagonal grid cells with 100 km spacings (figure 3*a,b*), using the R package dggridR [29]. Spatial points used in our spatial subsampling algorithm are therefore 100 km grid cells containing at least one fossil occurrence.

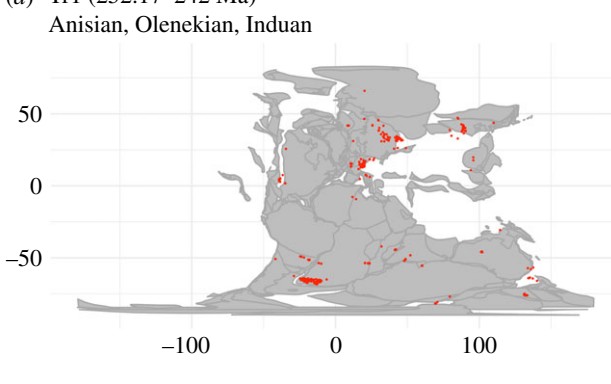

(a) Tr1 (252.17–242 Ma)
Anisian, Olenekian, Induan

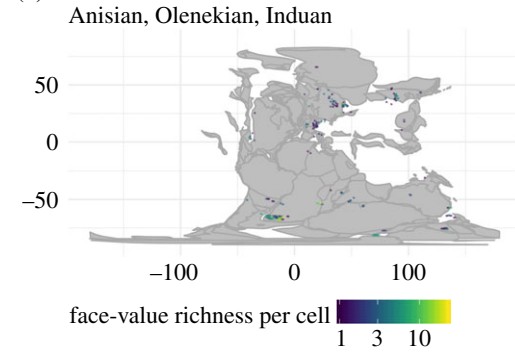

(b) Tr1 (252.17–242 Ma)
Anisian, Olenekian, Induan

face-value richness per cell
1   3   10

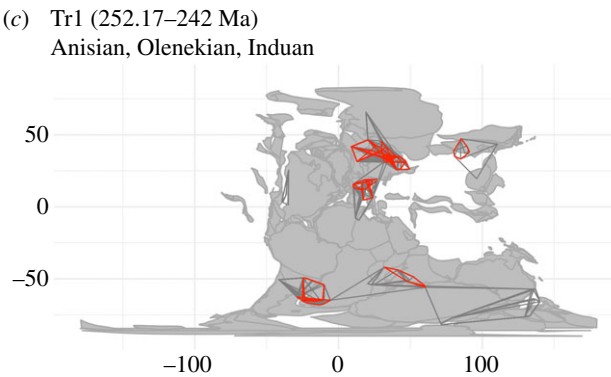

(c) Tr1 (252.17–242 Ma)
Anisian, Olenekian, Induan

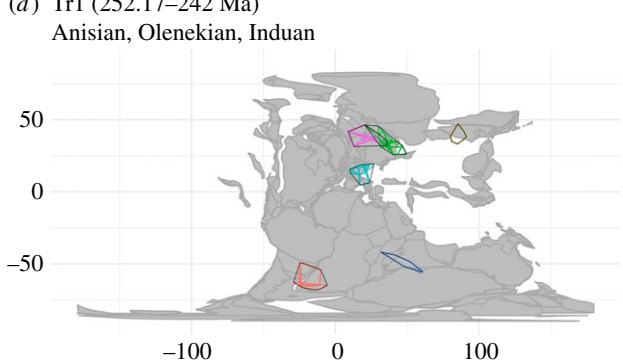

(d) Tr1 (252.17–242 Ma)
Anisian, Olenekian, Induan

**Figure 3.** Key steps in the spatial standardization procedure used in this study, showing samples for the Early–Middle Triassic (Tr1 time bin). (a) Palaeocoordinates of fossil localities. (b) Fossil localities binned within 100 km equal-size hexagonal/pentagonal grid cells (using dggridR). (c) Palaeogeographic regions delineated using convex hulls, with samples meeting spatial standardization criteria for 2000 km MST distance highlighted in red. (d) Clusters of highly similar palaeogeographical regions.

The spatial subsampling algorithm works by: (i) selecting a random spatial point as a starting location; (ii) identifying the closest spatial point, choosing at random if there are two or more equidistant points; (iii) saving these two points as a palaeogeographical region; (iv) identifying the closest point to those two points; (v) saving this set as a palaeogeographical region; and (vi) continuing this procedure until all spatial points have been added. The algorithm is then repeated for every possible starting location, and any duplicate palaeogeographical regions are discarded. Distances were calculated from midpoints of 100 km dggridR cells. This procedure results in a database of palaeogeographical regions (sets of directly adjacent or nearest-neighbour fossil localities) covering all possible sizes (figure 3c).

Palaeogeographical regions were identified using the set of fossil localities for the most inclusive taxon set that we analysed (i.e. non-flying, non-marine tetrapods). Diversity estimates for individual tetrapod subclades were also derived from these same palaeogeographical regions, because these represent areas in which tetrapod subclades could potentially be sampled.

Each palaeogeographical region was then characterized by computing a wide range of different metadata (e.g. variables relating to diversity, spatial factors or sampling metrics). Spatially standardized sets of palaeogeographical regions were obtained by simultaneously applying sets of filtering criteria (e.g. relating to spatial extent, numbers of occupied grid cells, etc.; see below).

## (d) Variables calculated for subsampled palaeogeographical regions

We calculated a wide variety of metadata for each palaeogeographical region. Spatial variables include the counts of occupied equal-area grid cells (i.e. cells yielding fossil occurrences) spanning a range of sizes (100, 200, 500, 1000 and 5000 km spacings, calculated using the R package dggridR [29]); our primary measure of palaeogeographical spread, MST length (= the minimum total length of all the segments connecting spatial points in a region [30]; see Close *et al.* [11] for justification); the distance of the longest branch in each MST (used to identify spatial regions with widely separated clusters of localities). Sampling variables include the counts of literature references reporting the fossil occurrences in each spatial region (used as a proxy for research effort) and measures of sample coverage (Good's $u$ [31] and the multiton ratio [32]).

We estimated species richness within palaeogeographical regions using four very different methods: face-value counts of species within regions (= raw or uncorrected richness; i.e. not sampling standardized), shareholder quorum subsampling (SQS [17,30,33], also known as coverage-based rarefaction [18]) and the asymptotic extrapolators 'squares' [34] and Chao 2 [35].

We focus primarily on patterns estimated using SQS, which provides an objective, frequency-dependent measure of diversity that is insensitive to variation in sampling [18]. Standardizing to equal sample coverage may increase the signal of evenness at lower quorum levels [20]. Nonparametric asymptotic richness extrapolators, on the other hand, are less sensitive to evenness, but are downward biased when sample sizes are insufficient for estimates to have asymptoted [20]. We therefore present estimates using both approaches. Face-value counts of species within palaeogeographical regions, meanwhile, facilitate direct comparison with existing face-value 'global' curves.

We implemented SQS using the analytical solutions in the R package iNEXT [36]), which allows seamless integration of interpolated (=subsampled), observed and extrapolated coverage-standardized species richness estimates. We used quorum levels of 0.4, 0.6 and 0.8.

## (e) Grid cell rarefaction algorithm

To additionally control for variation in the 'packing density' or spatial coverage of fossil localities within equal-sized palaeogeographical

regions, we used a grid cell rarefaction (GCR) procedure prior to calculating our focal measure of diversity, SQS (other estimators were not subject to this procedure owing to heavy computational demands). When using GCR, SQS was estimated for each palaeogeographical region at a range of subsampled grid cell quotas (we present GCR results using quotas of 3, 5 and 8 occupied 200 km equal-area grid-cells with per 1000 km of MST length, calculated using 50 subsampling trials). SQS richness was also estimated without GCR (GCR = 'off'). To compare different richness estimators on an equal footing, our focal results do not use SQS with GCR.

## (f) Standardizing spatial sampling

To standardize spatial sampling, we identified subsampled palaeogeographical regions that simultaneously met the following criteria:

(i) seven distinct spatial scales, comprising MST lengths of 1000 km, 1500 km, 2000 km, 2500 km, 3000 km, 3500 km and 4000 km (±10%; figure 3c and electronic supplementary material, figure S3). We quantified palaeogeographical spread using MSTs for reasons outlined by Close *et al.* [11];

(ii) MSTs for which the length of the longest branch was no more than 40% of the total MST size (in order to exclude clusters of localities separated by large gaps);

(iii) at least 20 literature references, to ensure a minimum level of study; and

(iv) a multiton ratio [32] of at least 0.25, to exclude palaeogeographical regions with very poor sample completeness (sometimes estimates of Good's $u$ may spuriously appear high for small sample sizes, and the multiton ratio offers a more conservative and partially independent measure of sample completeness).

We also excluded palaeogeographical regions that crossed geographical barriers, based on the combined presence of countries or continental regions at particular points in time (South America and Africa after 120 Ma; Australia and New Zealand after 70 Ma; Europe and Africa after 66 Ma).

## (g) Spatial clustering algorithm

Because our spatial subsampling algorithm finds all nested sets of adjacent spatial points, the full set of palaeogeographical regions will invariably include some regions that share underlying spatial points to a greater or lesser degree (ranging from no overlap to almost complete overlap). To address potential issues with non-independence between data points inflating apparent sample size, we identified clusters of similar palaeogeographical regions based on the fraction of spatial points they shared (samples were added to a spatial cluster if they shared greater than 25% of the spatial points with another sample in the cluster; figure 3d and electronic supplementary material, figure S4). Key variables such as diversity and spatial or sampling metrics were then summarized for each cluster of palaeogeographical regions by computing median values and interquartile ranges.

## (h) Model comparisons

We used linear model comparisons to examine whether patterns of spatially standardized diversity are more consistent with diversification that is unconstrained ('expansionist', with steady increases through time) or constrained (i.e. with long-lived diversity equilibria, separated by phase-shifts). Our linear models included combinations of three explanatory variables: (i) absolute time, representing continuous per-lineage diversification; (ii) an intercept, representing a null model in which diversity is static through time; and (iii) a diversification-phase variable in which the intercept and/or slope are allowed to differ before and after the Cretaceous/Palaeogene (K/Pg) mass extinction (66 Ma).

Phase was included both as a covariate (allowing the intercept to vary independently between phases) and an interaction term (allowing the intercept and slope to vary between phases; see the electronic supplementary material, table S2 for full list of models). These models were compared against an intercept-only null model. Richness estimates were log-transformed. Models were ranked using Akaike information criteria with the adjustment for small sample sizes (AICc) [37].

## (i) Interactive data explorer

Patterns of spatially standardized diversity and other variables can be explored interactively using an RStudio Shiny app included in the Dryad Digital Repository (by following the instructions provided in the file named README.md [27]). The interactive data explorer app allows exploration of spatially standardized diversity results for all taxon sets, richness and other variables. Clicking on a data point plots the underlying data on a palaeomap and displays tables of the underlying occurrence data in that palaeogeographical region.

# 3. Results

The palaeogeographical spread (=spatial extent) of the terrestrial fossil record grows exponentially through the Phanerozoic (figure 1b and figure 2; see also the electronic supplementary material, figures S1 and S2) and is decoupled from the actual terrestrial area that existed through time. Although the palaeogeographical spread of the sampled fossil record increases fourfold through the Cenozoic, increases in actual terrestrial area over the same interval are much smaller (approx. 15%; [19]; figure 1b; electronic supplementary material, figure S5). Changes (i.e. first differences) in the palaeogeographical extent of the 'global' fossil record of terrestrial tetrapods explain approximately 24–67% of changes in face-value species counts, and 31–34% of the changes in subsampled richness estimates, depending on the measure of palaeogeographical spread used (figure 1c,d and electronic supplementary material, figure S6). By contrast, changes in the palaeogeographical spread of the fossil record are not significantly correlated with changes in continental area (figure 1e,f and electronic supplementary material, figure S7). The strong correlations observed between diversity and spatial sampling therefore represent real and profound sampling biases [10,11,20,24] that cannot be explained by 'redundancy' or 'common cause' effects [38,39].

The non-marine sedimentary rock record also decays exponentially with increasing age owing to the progressive loss of sediments to erosion and burial and is therefore likely to exert some influence on the palaeogeographical spread of fossil localities through time [16,40,41]. Surprisingly, though, we find that neither changes in 'global' diversity nor the palaeogeographical spread of the fossil record are significantly correlated with changes in extent of non-marine sediments (electronic supplementary material, figure S8). This indicates that the rock record is not the primary factor controlling spatial sampling in the terrestrial fossil record and further justifies our direct use of the palaeogeographical distribution of the tetrapod fossil record to estimate spatially standardized diversity patterns. Generalized least-squares models (GLS) of 'global' diversity, as a function of the palaeogeographical spread of the worldwide fossil record, continental area and non-marine sediment extent (modelling temporal autocorrelation using a first-order autoregressive structure), recover a strong, statistically significant explanatory role only for palaeogeographical spread (electronic supplementary material, table S3).

Because pervasive spatial bias prevents us from estimating meaningful time series of global diversity through the Phanerozoic, we recommend that studies must instead focus on estimating regional-scale diversity for well-sampled palaeogeographical regions. The patterns of spatially standardized regional richness that we recover are broadly consistent across spatial scales and for different richness estimators (figure 4). Surprisingly, results are highly congruent even when using face-value counts of species from spatially standardized regions (in other words, when spatial sampling is standardized, but sampling intensity is not; figure 4). This suggests that variation in the spatial scope of the terrestrial fossil record has a more pronounced effect on apparent species richness than does variation in intensity or completeness of sampling within those regions.

Although data are insufficient to estimate regional diversity for much of the Palaeozoic, levels during the latest Permian (approx. 255 Ma) appear to have been similar to those of the Early Triassic (approx. 250 Ma; figure 4). Similar regional diversity estimates are maintained up until the latest Cretaceous (approx. 70 Ma), spanning a total interval exceeding 180 Myr. Linear regressions of diversity on time for this extended interval return non-significant slopes, indicating a long-term static pattern of standing regional diversity (electronic supplementary material, figure S9). This is true despite substantial faunal turnover throughout, including the Permian/Triassic (P/T) mass extinction (252 Ma), and the initial origins of groups that are speciose today during the Jurassic and Cretaceous [42].

Nevertheless, there are two clear intervals when regional-scale tetrapod diversity apparently increased substantially. All tetrapods share a single ancestor species that lived no later than the Late Devonian [43]. Although the data are insufficient to obtain diversity estimates during the Carboniferous, early increases in terrestrial tetrapod diversity must therefore have occurred within the Carboniferous to mid-Permian. A large apparent increase in maximum regional diversity also occurred later, in the aftermath of the K/Pg mass extinction [10,11,24]. This primarily results from the fossil record of mammals, which shows an abrupt three- to fourfold increase in regional diversity (figure 5). There is no evidence in our data for substantial increases in maximum regional diversity through the remainder of the Cenozoic, either in tetrapods as a whole, or in major subclades (figures 4 and 5). In fact, the linear regressions of regional diversity on time for the Cenozoic recover significant trends towards lower richness through time, driven by lower diversity in bins Ng3 and Ng4 (approx. the last 10 Myr; electronic supplementary material, figure S9).

Model selection using information criteria demonstrates that the best explanations of regional diversity include the passage of time and a phase-shift across the K/Pg boundary. Across all spatial standardization criteria, the model including time and phase as an interaction term receives greatest support (electronic supplementary material, table S2). This is because there is a shift to a higher regional diversity equilibrium across the K/Pg boundary, but this is followed by a significant decrease in regional diversity towards the present (electronic supplementary material, table S4 and figure S9). For other richness estimators, see the electronic supplementary material Results.

GCR results highlight that the density of spatial coverage inside standardized palaeogeographical regions increases towards the present: when higher quotas of occupied grid cells are imposed, many more data points are excluded from the Palaeozoic–Mesozoic than from the Cenozoic (electronic supplementary material, figure S10).

## 4. Discussion

Although long under-appreciated, variable spatial sampling represents a fundamental fossil record bias, and one that must be accounted for. Our results show that previous interpretations of exponential increases in tetrapod diversity through the Phanerozoic are an artefact of the increasing spatial extent of the 'global' fossil record (figure 1a,b). Between one and two thirds of the changes through time seen in 'global' diversity curves can be explained by changes in the palaeogeographical extent of sampled fossil localities (figure 1b,d and electronic supplementary material, figure S6), and this covariation is not explained by changes in the actual amount of habitable land area (electronic supplementary material, figure S7E–H) or the extent of non-marine sediments (electronic supplementary material, figure S8F–J). Although changes in continental area and the extent of non-marine sediments through time probably do exert some influence on the worldwide palaeogeographical spread of the terrestrial fossil record (particularly the extent of non-marine sediments, which decreases exponentially with increasing age [41]), other factors appear to be at least as important.

Estimating truly representative 'global' diversity curves for terrestrial tetrapods is, therefore, almost certainly not possible based on our current knowledge of the fossil record, and diversity analyses must focus on local and regional scales. We present, to our knowledge, the first spatially standardized regional richness estimates spanning the entire evolutionary history of tetrapods. By estimating diversity for comparably sized palaeogeographical regions through time, we recover fundamentally different patterns of diversity change to those found by previous studies of face-value 'global' trends [5,9], even when we consider only face-value species counts that do not control for variation in sampling intensity (figure 4). Most notably, variation in regional diversity within individual time bins is usually on par with variation through time, leading to patterns that are constrained over timescales of up to approximately 180 Myr. We find no support for large sustained increases over the last 100 Myr.

We do, however, observe an abrupt increase in regional-scale terrestrial tetrapod diversity during the earliest Cenozoic, consistent with recent work at local to continental spatial scales [10,11,24]. The precise reasons for this step-change are currently uncertain. It may support a fundamental role for the K/Pg mass extinction in disrupting and reorganizing terrestrial ecosystems, consistent with a role for ecological limits in regulating diversification [4]. Mammals certainly experienced a large increase in richness in the early Cenozoic. However, the relative contribution of mammals to overall tetrapod diversity patterns—and thus the magnitude of the increase itself—is probably exaggerated, owing to their high preservation potential and the ease of diagnosing species from isolated teeth: in the Cenozoic fossil record, mammal diversity is more than twice that of squamates (figure 5), yet the reverse is true for extant species richness. By contrast, the P/T extinction, the largest in Earth history, does not at present appear to have played a similar role in elevating long-term diversity (although sparse Palaeozoic data limits interpretations). The reasons for the

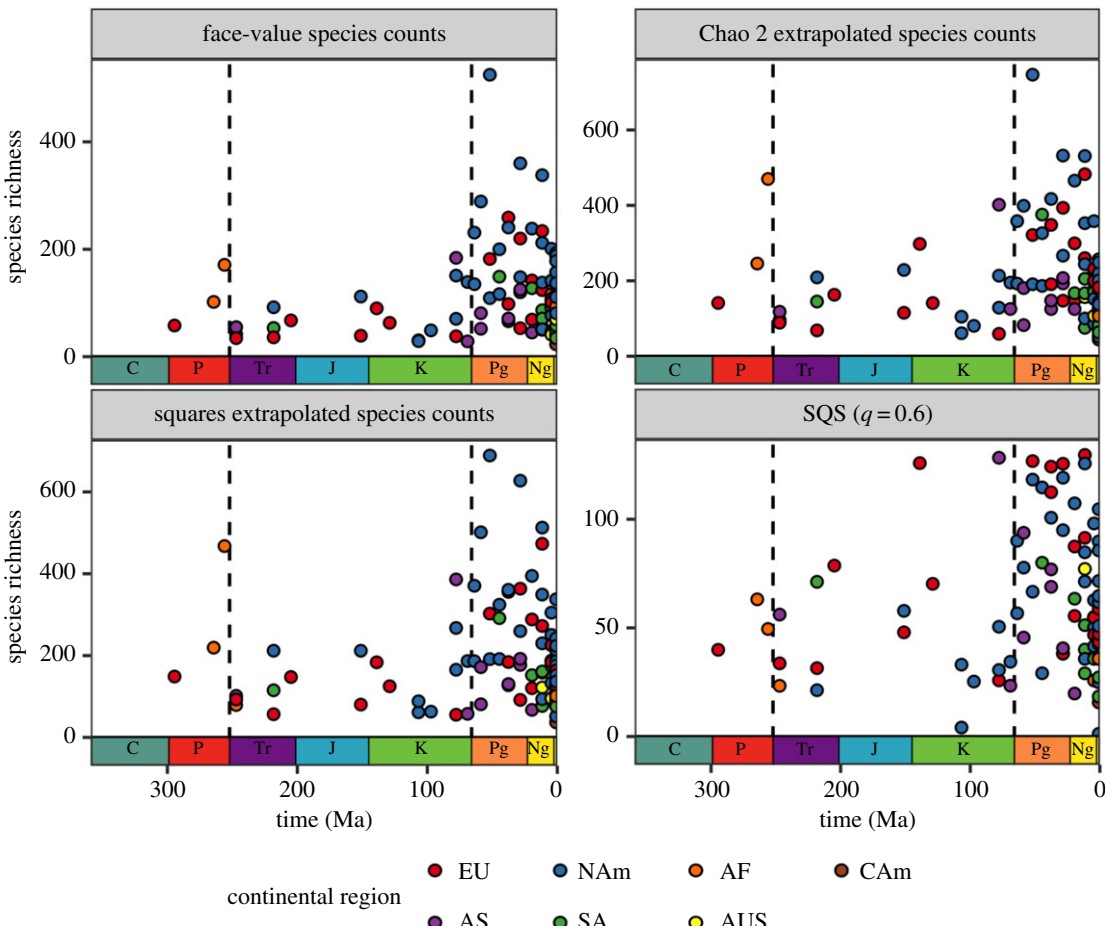

**Figure 4.** Patterns of spatially standardized regional-scale species richness of non-flying terrestrial tetrapods through the Phanerozoic, for regions 2000 km in size (minimum-spanning tree (MST) distance). Patterns depicted using face-value (but spatially standardized) species counts, squares [34] and Chao 2 extrapolated richness [35], and SQS [17,18] (using quorum = 0.6). Grid cell rarefaction algorithm not used (GCR = off). Colours correspond to dominant continental regions of palaeogeographical regions. Data points represent median richness estimates for clustered palaeogeographical regions. NAm, North America; EU, Europe; SA, South America, AF, Africa; AS, Asia, AUS, Australasia; CAm, Central America.

differing long-term impacts of the P/T and K/Pg extinctions on standing terrestrial diversity are unclear, but may reflect differences in the timescales over which the two events took place, or variation in the biology and preservation potential of the groups that flourished in the aftermath of each event.

Meanwhile, we find no evidence for effects on regional diversity of other events in the evolutionary history of terrestrial tetrapods that have been hypothesized to have catalysed diversity increases, including the initial expansion of angiosperms during the middle and Late Cretaceous [7], and the breakup of the supercontinent Pangea [44]. This does not rule out a role for events in plant evolution as drivers of tetrapod diversification. Instead, it is possible that floral state-changes across the K/Pg boundary (e.g. increases in seed sizes [45]) might have been more important for mammalian species richness than events within the Cretaceous itself, a hypothesis that requires further investigation. Neither do our analyses of regional diversity rule out some increase in global richness owing to continental fragmentation (although we have shown that global diversity cannot currently be directly estimated). Modelling of species–area relationships suggests that this effect could have approximately doubled global terrestrial tetrapod biodiversity between the Triassic and Late Cretaceous, during the main interval of Pangean fragmentation [44]. Pangean fragmentation was largely complete by the end of the Cretaceous, and it seems unlikely that the comparatively minor continental rearrangements that occurred during the Cenozoic could have driven the

proposed tenfold increase in global diversity recovered by influential previous work [5,9].

Our results are consistent with a growing body of evidence from the fossil record for constrained diversification within the terrestrial realm [6,10,11,21,24,34,46,47]. Moreover, the regional-scale patterns we document for Phanerozoic tetrapods are highly congruent with those observed at smaller spatial scales, such as for local richness [24], which also show minimal increases from the late Palaeozoic–Mesozoic, a step-change across the K/Pg boundary, and no increase through the Cenozoic. The similarity between patterns of diversity at local (alpha) and regional (gamma) scales suggests an absence of systematic long-term trends in tetrapod beta diversity within regions through the Phanerozoic, although studies of the long-term patterns of beta diversity are needed to confirm this. Although limitations of the fossil record prohibit us from analysing regional-scale flying tetrapod diversity here, within-community patterns suggest these groups (birds, bats and pterosaurs) were also subject to long-term constraints [24]. These patterns suggest that the early diversification of birds resulted in the stepwise addition of substantial species richness to terrestrial ecosystems [10], with limited subsequent increases [24] that mirror the patterns of tetrapod richness documented here.

The diversity patterns we present are for regional spatial scales and thus not directly comparable with global patterns. Furthermore, our results suggest that truly global estimates of tetrapod diversity through geological time are inaccessible based

*Proc. R. Soc. B* **287**: 20200372

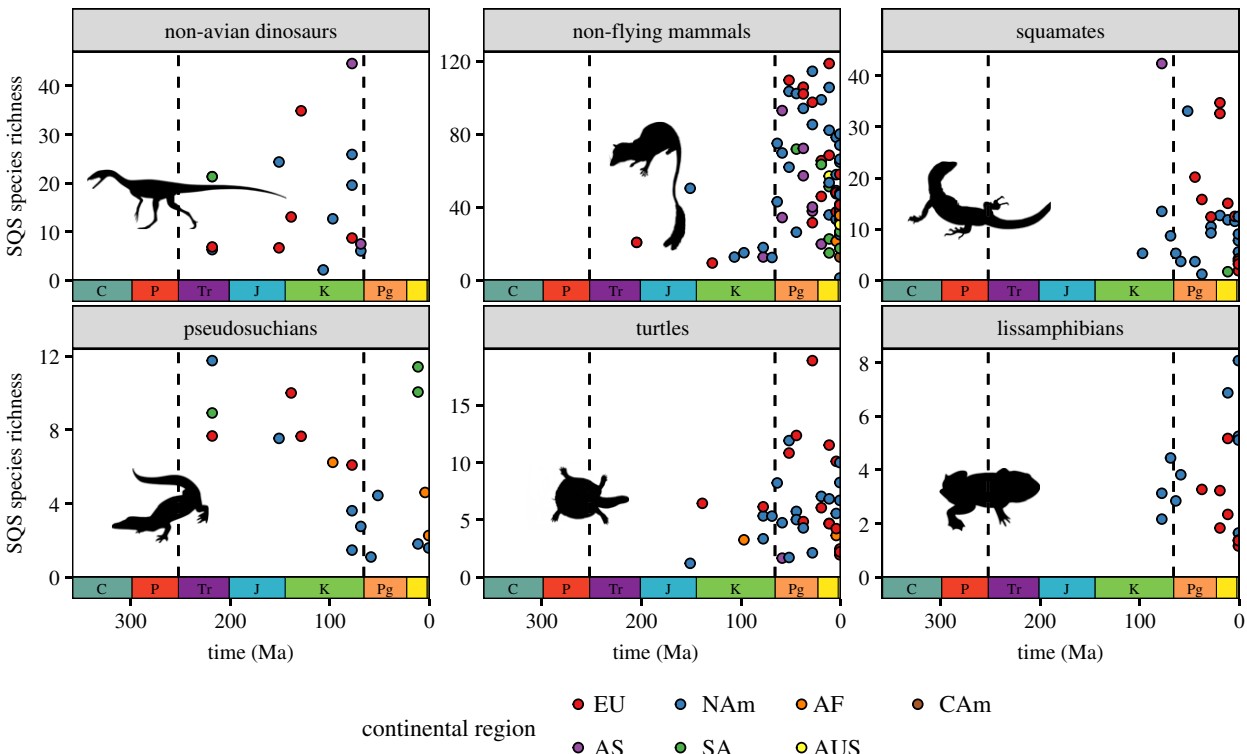

**Figure 5.** Patterns of spatially standardized regional-scale species richness for major subclades of non-flying terrestrial tetrapods (non-avian dinosaurs, non-flying mammals, squamates, pseudosuchians, turtles and lissamphibians), for regions 2000 km in size (minimum-spanning tree (MST) distance). Species richness estimates extrapolated using SQS (quorum = 0.6, GCR = off). Colours represent dominant continental regions of palaeogeographical regions. Silhouettes courtesy of Phylopic (http://www.phylopic.org). Image credits for Phylopic silhouettes: non-avian dinosaur by Ian Reid, CC BY-NC-SA 3.0; non-flying mammal by FunkMonk/Michael B. H. (CC BY-NC-SA 3.0); squamate by Ghedo and T. Michael Keesey (CC BY-SA 3.0); pseudosuchian by Phylopic (Public Domain Mark 1.0); turtle by Phylopic (Public Domain Dedication 1.0); lissamphibian by Nobu Tamura (CC BY 3.0).

on our current knowledge of the fossil record. Nevertheless, barring substantial and as-yet-unquantified increases in global-scale faunal provinciality (i.e. between continental regions), previous findings of sustained, expansionist increases in 'global' standing diversity over the last 100 Myr [5,7,9] are most likely artefactual, resulting from a failure to account for exponential increases in the spatial extent of terrestrial sampling over the same interval. Our results provide further evidence to overturn the previous paradigm of unconstrained, expansionist diversification, instead indicating long periods of relative stasis, disrupted by rare, geologically rapid rises in maximum standing diversity.

**Data accessibility.** Data is available from the Dryad Digital Repository: https://doi.org/10.5061/dryad.280gb5mmv [27].

**Authors' contributions.** R.J.B. and R.A.C. conceived the study. J.A., M.T.C., M.D.U., P.D.M., R.J.B., R.B.J.B., T.J.C. and E.M.D. contributed to the dataset. R.A.C. designed and conducted the analyses, and made the figures. R.B.J.B. and R.J.B. provided methodological input. R.J.B. and R.A.C. wrote the manuscript. All authors provided critical feedback on the text.

**Competing interests.** We declare we have no competing interests.

**Funding.** This research was funded by the European Union's Horizon 2020 research and innovation programme under grant no. 637483 (ERC Starting Grant TERRA to R.J.B.). P.D.M. was supported by a Royal Society University Research Fellowship (UF160216).

**Acknowledgments.** We thank all contributors to the Paleobiology Database. This is Paleobiology Database official publication 363. We thank Mike Benton and two anonymous reviewers for helpful review comments.

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
