## [Reviewer comments · Proceedings of the Royal Society B: Biological Sciences]

Review History

RSPB-2019-2254.R0 (Original submission)

Review form: Reviewer 1 (Michael J. Benton)

Recommendation

Major revision is needed (please make suggestions in comments)

Scientific importance: Is the manuscript an original and important contribution to its field?

Good

General interest: Is the paper of sufficient general interest?

Good

Quality of the paper: Is the overall quality of the paper suitable?

Good

Is the length of the paper justified?

Yes

Should the paper be seen by a specialist statistical reviewer?

Yes

Do you have any concerns about statistical analyses in this paper? If so, please specify them explicitly in your report.

No

It is a condition of publication that authors make their supporting data, code and materials available - either as supplementary material or hosted in an external repository. Please rate, if applicable, the supporting data on the following criteria.

Is it accessible?

Yes

Is it clear?

Yes

Is it adequate?

Yes

Do you have any ethical concerns with this paper?

No

Comments to the Author

This contribution does two novel things. First, it provides a set of palaeodiversity curves for terrestrial tetrapods based on current data, as downloaded from the Paleobiology Database, and (2) it 'corrects' these data by paying attention to what the authors call 'spatial bias'. This project forms part of a long-running debate among palaeobiologists, that was started by Raup's influential 1972 paper in *Science* in which he suggested that the 'fossil record' might be more bias than signal. His argument was that our knowledge decays backwards in time and so the apparent rise of palaeodiversity from the dim distant past to the present day could simply record those biases (geological and human). What if, he asked, global diversity had always been more or less constant, and we simply sample an increasing amount of it from the Cambrian to the present?

In their study, the current authors sampled all tetrapods, and removed marine forms (ichthyosaurs, plesiosaurs, whales etc) and flying forms (primarily pterosaurs, birds and birds). This is fine. The plots of raw data and sampling-standardised data in Figure 1c appear to show the classic somewhat exponential curves, with very low levels of diversity through the late Palaeozoic and Mesozoic, and a substantial uplift through the Late Cretaceous and Cenozoic to reach levels 3–4 times Late Cretaceous levels in the Neogene/ Quaternary.

The key contribution here is to apply corrections for 'spatial bias', an idea posited before by Close et al. (2017). This controls for variation in the palaeogeographic extent of fossil record sampling within intervals. Basically, their estimate of habitable area of the land surface (green curve in Fig. 1a) tracks the curves for palaeodiversity (Fig. 1c, both curves), and if one accepts that the amount of habitable area drives recorded diversity, then it makes sense to correct the one by the other and achieve a more or less flat line. Hence, they conclude that diversity of tetrapods has not risen exponentially through time, or at least through the last 100 Myr, and that global diversity was constrained over spans of tens or hundreds of millions of years, for all tetrapods, and for subsets of tetrapods.

So, if their correction method based on habitable areas is valid, they are right; if not, their suggestion contributes nothing.

Figure 1A shows that habitable area increases by maybe 33% (20% to 30% of Earth's surface), whereas their metric of spatial sampling increases from 0 to 100% along a curve that is very close to that of uncorrected global species richness (Fig. 1B). Such close tracking suggests that the two signals are linked, but which drives which? It could be one of two explanations:

1. Spatial sampling drives apparent palaeodiversity, and so the latter should be corrected by the former, as it is a sampling signal.
2. Spatial sampling reflects actual increases in palaeodiversity, and so is a redundant signal with raw diversity.

As an initial observation, these close correlations used to be accepted as sure evidence that, for example, numbers of specimens, collections or localities provided a valid proxy for sampling, but their close tracking of the raw palaeodiversity signal on the other hand really reflected redundancy, as I argued many times, and as these authors now seem to accept.

Now, I know that spatial sampling is a real thing, and we demonstrated it in our study of the British fossil record (Dunhill et al. 2014), but I hope the authors will accept that everything they say is not self-evident, and it is worth taking a critical look at what they are doing. Can they categorically rule out that for some or all clades, there might have been times when real distributions were patchy or limited to certain continental areas? In other words, since the Carboniferous, the ways in which the surface of the Earth was habitable have varied. Three things:

1. In the Carboniferous, perhaps tetrapods occupied only 10% of the apparently 'habitable' land surface because most were tied to freshwaters (temnospondyls, anthracosaurs, lepospondyls), and the areas covered by plants and soils perhaps extended only a few 100 m away from water courses, leaving hills, mountains and centres of continents as bare rock. Very likely continental centres remained barren through the Permian and Triassic as well – that could be >50% of the Earth devoid of tetrapod life in reality. Therefore, correcting the raw diversity values (whether for continents or the world) by assuming those areas ought to have been habitable according to modern standards would artificially inflate the total diversity estimates, perhaps by 100% or more.
2. Pangaea split in stages and the degree of endemism of terrestrial tetrapods has perhaps changed through time. Some might say this has been a big influence on diversification at various levels; others might say it has not been. But, again, applying a method that takes account only of total land area of particular regions or the world might miss any impact of the combined continental position/ sea level on endemism and hence total global diversity.
3. Perhaps climatic changes coupled with changing physiologies would have other non-uniform effects. As you generate climate belts, you generate diversity of habitats (maybe an effect of doubling or trebling the diversity of habitat types from say Jurassic to Neogene), and this might have different effects on endotherms and ectotherms and their ability to generate diversity. Is this factored into the spatial metrics?

I know this is repeating stuff from Sahney et al. (2010) and elsewhere, but in the context of application of a single method to all data through a massively changing world might give some pause for thought.

Two further thoughts:

1. By excluding birds, you remove 10,000 modern species, a fair chunk of modern tetrapod biodiversity, but I can follow the reasoning in terms of sampling.
2. More fundamentally though is that the study makes its very large claims based only on fossil data, in other words clipping the raw global diversity at the modern end of the time scale to maybe 1100 species, instead of the actual 30,000 species (or 20,000 species without birds). This means, I assume, what you are saying is that 'corrected' global diversity for the Cenozoic is perhaps 20,000 species throughout, and half that, 10,000 species, throughout the whole of the Mesozoic.

Line 47: I'm not sure these studies were based on 30-year-old compilations. We used the Fossil Record 2 (1993) in the 1996 paper, which is now 26 years old, but was only 3 years old then (ref. 4). For Sahney et al. (2010, ref. 8) these data were then 17 years old.

Lines 265-266: 'Our results are consistent with a growing body of evidence [5,9,10,15,16,28,33,34] for constrained diversification within the terrestrial realm.' Interesting – I suspect a lot of terrestrial ecologists and phylogenomics people will be a bit startled. Maybe you could explain what these constraints are that mean changing geographies through the Cenozoic have had negligible effect on summed global diversity. Also, perhaps a word about the apparent explosion in diversity of clades like murid rodents (or passeriform birds – but birds are excluded) – they were presumably then replacing a precursor equally diverse clade that disappeared or shrank? There can be no suggestion in a world of constraint such as you propose that such clades could actually have hit on a smart new mode of life and that they really did diversify explosively and without displacing pre-existing species.

And maybe a word to those who are interested in gymnosperms and angiosperms – in the modern world, angiosperm-dominated forests tend to be rich in life of all sorts, whereas gymnosperm-dominated forests are not. In terms of animal diversity this can be an effect of one or more orders of magnitude. Perhaps the switch from one to the other through the Cretaceous did nothing to insects, lizards, mammals, etc? I certainly can't prove the shift in dominant plants did drive diversity up, but many would accept that likely this did, and it might be good if you could explain why in fact the Cretaceous angiosperm expansion made no difference to global diversity of tetrapods.

Mike Benton

Review form: Reviewer 2

Recommendation

Accept with minor revision (please list in comments)

Scientific importance: Is the manuscript an original and important contribution to its field?

Excellent

General interest: Is the paper of sufficient general interest?

Excellent

Quality of the paper: Is the overall quality of the paper suitable?

Excellent

Is the length of the paper justified?

Yes

Should the paper be seen by a specialist statistical reviewer?

No

Do you have any concerns about statistical analyses in this paper? If so, please specify them explicitly in your report.

No

It is a condition of publication that authors make their supporting data, code and materials available - either as supplementary material or hosted in an external repository. Please rate, if applicable, the supporting data on the following criteria.

Is it accessible?

Yes

Is it clear?

Yes

Is it adequate?

Yes

Do you have any ethical concerns with this paper?

No

Comments to the Author

This paper concerns the pattern of tetrapod species richness across the last c. 300 million years, and more particularly whether it fits an expansionist or constrained model of diversification. The methods used are appropriate and cutting edge and further emphasise the importance of accounting for spatial factors in palaeontological estimates of taxonomic richness. Importantly, the results strongly favour the constrained hypothesis, aka, "diversity dependence" or a "carrying capacity" model, with an upwards shift in this value coincident with the K-Pg boundary.

I am happy to support publication almost as is, except for a few minor corrections below. Aside from this, I personally think the manuscript could be improved by considering what we might expect the pattern of tetrapod diversity to be over the sampling window. For example, major innovations we might expect to increase richness are: 1) the origin of tetrapod herbivory, 2) the origin of flowering plants and, 3) the expansion of grasslands. None of these are coincident with the one shift the authors favour (at the K-Pg). Additionally, two major geographic considerations seem undiscussed. Firstly, the effects of the assembly and particularly the breakup of the Pangaeian supercontinent on richness has been much discussed in the literature (e.g., Jordan et al. 2016, *Phil. Trans. Roy. Soc. B.*, 371, 20150221; Vavrek 2016, *Biol. Lett.*, 12, 20160528). Both of those papers seemingly make predictions that are not supported by the results presented here and should perhaps be used as additional context for interpretation. Secondly, the authors' results appear to rule out a species-area effect in tetrapod biogeography, something that studies of extant richness typically support. This seems like an additional and major implication of the results that the authors do not currently discuss.

Line-by-line comments:

L82 - "All data downloads were performed in the analysis R scripts" Grammatically this made no sense to me. Suggest rewording.

L83 - "were recorded as text files deposited on Dryad" Missing "and"?

L88 - "comprised 10,699 collections (= localities)" I am not sure this is a safe assumption (i.e., collections do not always perfectly represent localities). However, the authors have applied spatial grouping methods that account for this issue, so perhaps this could be stated instead.

L91 - "if over 50% of their temporal duration was contained" Somewhat pedantic, but do tetrapod occurrences truly have a duration? I would have thought this really means temporal error. A single fossil has no duration at all. Suggest rewording to clarify. A deeper question might be what this scheme means for temporal patterns. I.e., does it "clump" occurrences, "smooth" them etc. What about very poorly temporally constrained occurrences? And perhaps ultimately, what proportion of the data does this represent?

L112-115 - The numbering here is incorrect (3 is repeated).

L118 - Equal-size, but what shape are these? A grid implies rectangles, but obviously the curvature of the surface of a sphere significantly complicates this. Additionally, one imagines membership of a grid cell is dependent on the "anchoring" of this grid as any covering of a sphere is essentially infinitely rotatable. What was the chosen origin? Or, most simply, can a figure showing this grid on a map be included in the SI.

L133 - "in order to excluding" Grammar.

L257 - Alternatively, perhaps mammals have been oversplit? Or this is an issue with temporal binning, time-averaging higher-temporal resolution collections of short duration taxa together? I

do not think it is on the authors to solve this question, but this seems like an obvious "further work" part of the discussion.

L259 - "Palaeozoic" should be "Paleozoic". This is a formal name not a UK- vs US-spelling issue.

L384 - "500 km spacings", earlier (L118) 100 km spacings were stated and in the SI 200 km. Some clarity would help here.

L402 - Assuming these are Phylopic silhouettes the artists should be given specific credit and some may require permission to use. Please check the web site: <http://phylopic.org/about>.

Review form: Reviewer 3

Recommendation

Accept with minor revision (please list in comments)

Scientific importance: Is the manuscript an original and important contribution to its field?

Good

General interest: Is the paper of sufficient general interest?

Excellent

Quality of the paper: Is the overall quality of the paper suitable?

Excellent

Is the length of the paper justified?

Yes

Should the paper be seen by a specialist statistical reviewer?

Yes

Do you have any concerns about statistical analyses in this paper? If so, please specify them explicitly in your report.

No

It is a condition of publication that authors make their supporting data, code and materials available - either as supplementary material or hosted in an external repository. Please rate, if applicable, the supporting data on the following criteria.

Is it accessible?

Yes

Is it clear?

Yes

Is it adequate?

Yes

Do you have any ethical concerns with this paper?

No

Comments to the Author

Overall this is an excellent paper, the analyses are well thought out and executed and the manuscript is clearly written. I don't have methodological or inference concerns. I have some minor comments in the attached pdf on the writing in the methods section. I have no comments

on the supplementary information. The online tool is a very useful addition that helps in understanding the analyses and patterns.

I have one specific thing to highlight from the comments in the pdf. The analyses in this paper are complex with many steps that become hard to keep track of, I think it would be very useful to have a methods figure with a schematic of the procedure to make it more easily understandable. A flow chart of the pipeline, something like in Gilbert and Escarguel 2017 "Evaluating the accuracy of biodiversity changes through geologic times: from simulation to solution". Or even better explanatory images of toy examples to go with it. I think it's worth taking the time to explain things in straightforward terms to make it as readable (and therefore as citable) as possible given the potentially broad audience that might be interested in this.

One other question is which figure should be used for the main manuscript figure. Perhaps the authors feel it would be misleading to select one of the 12 boxes and present that (if so ok leave things as they are, or perhaps even just switch Fig 2 and fig S3), but I think if it is made clear in the caption that all the other approaches show broadly the same result I think that is ok. The paper presents a convincing 'global' terrestrial diversity plot for the Phanerozoic, I know its not possible to actually do a global curve because of the nature of the data and analyses but at the moment I think having 12 virtually identical plots in the main results figures buries the message and impact of the result. From my perspective the ideal way to make very clear the main result would be to select one of the plots from supplementary figure 3 (maybe squares 3000 or sqs 3000) that has the model lines on it as the main results figure then put the rest in the supplement, or have the current fig 2 as a fig 3.

Decision letter (RSPB-2019-2254.R0)

28-Nov-2019

Dear Dr Butler:

I am writing to inform you that your manuscript RSPB-2019-2254 entitled "The apparent exponential radiation of Phanerozoic land vertebrates reflects spatial sampling biases" has, in its current form, been rejected for publication in Proceedings B.

This action has been taken on the advice of referees, who have recommended that substantial revisions are necessary. With this in mind we would be happy to consider a resubmission, provided the comments of the referees are fully addressed. However please note that this is not a provisional acceptance. One reviewer had major reservations that will take some new analysis and convincing; others and the Associate Editor request pitching the paper's text better to the broad audience of Proc B.

- 1) A 'response to referees' document including details of how you have responded to the comments, and the adjustments you have made.

- 2) A clean copy of the manuscript and one with 'tracked changes' indicating your 'response to referees' comments document.
- 3) Line numbers in your main document.

Please note that this decision may (or may not) have taken into account confidential comments.

In your revision process, please take a second look at how open your science is; our policy is that all data involved with the study should be made openly accessible-- see: <https://royalsociety.org/journals/ethics-policies/data-sharing-mining/>
Insufficient sharing of data can delay or even cause rejection of a paper.

Sincerely,
Professor John Hutchinson, Editor
mailto: proceedingsb@royalsociety.org

Associate Editor
Comments to Author:

Thank you for the opportunity to review this very interesting paper. Herein the authors re-examine species-level diversity distributions in (most) terrestrial vertebrates through the last ~400 million years, taking into account spatial sampling bias for the first time in this context. They use this approach to examine the relative support for two competing models of diversity through geological time given the fossil data available: diversity expanded through time versus diversity remained more or less constant through time. The implication of the latter model is that the observed expansionist trends in raw data through time are in fact driven by spatial sampling bias in the fossil record, and thus not a true evolutionary signal. This is a very topical area and one likely to be of interest to a broad range of biological and geological scientists. All three reviewers are positive about the goals and approaches of the study, but have forwarded a number of criticisms and/or suggested improvements that should be made before publication. I find myself agreeing with all these suggestions, and perhaps even going a little further in some cases.

As a biologist working outside this immediate area of investigation, I found it extremely difficult to follow (even conceptually at times) the many, many steps in data processing and analysis that ultimately lead to spatially considered/corrected diversity analyses. This issue is raised by expert Reviewer 3 and I agree that at present there is little hope that the general reader of Proceedings B will be able to judge exactly what is being done, why it is being done and how valid or crucial certain assumptions are to the big picture. Reviewer 3 highlights some examples in their annotated pdf, and I would also highlight sentences like "Occurrences were assigned to a bin if over 50% of their temporal duration was contained within a single bin." Adding explanations of why these kinds of steps were done, and in other cases some evaluation of the assumptions underpinning such methodological decisions, would be very helpful to the readership. I would also like some critical thought or evaluation of raw data being used to quantify land area/spatial distributions in the fossil record. Surely these data are estimations and contain error and potentially even some systematic temporal bias in that error. Or not?

Reviewers 1 and 2 ask for a much broader and perhaps more balanced discussion of the results and have independently cited similar topics that warrant detailed treatment given the specific trends and overall qualitative interpretation favoured by the authors. Many of these things occurred to me when first reading the study, and in fact certain (almost throwaway) statements in the paper made me think even more critically than Reviewers 1 and 2 have about the take home

message carried by the paper. The paper is set up to evaluate the validity of the expansionist model of diversity. Having put forward evidence that the expansionist model is not supported, the authors favour a model of relative stasis in diversity through time. The authors are mostly (but perhaps not always) careful not to push the support for stasis model too hard, but ultimately I was left feeling that a third equally valid interpretation was not discussed at all in the paper (at least not explicitly): that the raw fossil (and land area?) data is fundamentally not good enough to pick up the true diversity signal, whatever it is. There's little if any explicit discussion of this in the paper, but surely it is fundamental to deciding between the competing models. I'm not suggesting that the authors somehow quantify exactly how incomplete the Phanerozoic fossil record actually is, but there are a number of their own points that, to me, make the conclusion that the raw fossil data isn't good enough to discern between competing models inescapable, at least as a possible conclusion. These include, for example, the significant impact that well-sampled mammal assemblages have on particular time bins, the fact that relative mammalian vs squamate diversity is reversed at certain points in recent history, and not least the statement in the methods that "Estimating truly representative 'global' diversity curves for terrestrial tetrapods is therefore not possible based on our current knowledge of the fossil record." It seems possible that the additional discussion asked for by Reviewers 1 and 2 may lead to a situation where "the data ultimately isn't good enough for us to tell" feels like an even more reasonable conclusion. Maybe I'm being overly pessimistic, or missing something fundamental, but I feel at the very least this conclusion should be explicitly discussed (and if valid to do so, objectively discounted) in the paper.

Given all this, it is my recommendation that the authors be encouraged to submit a revised version of the manuscript to Proceedings B for further consideration.

Reviewer(s)' Comments to Author:

Referee: 1

Comments to the Author(s)

This contribution does two novel things. First, it provides a set of palaeodiversity curves for terrestrial tetrapods based on current data, as downloaded from the Paleobiology Database, and (2) it 'corrects' these data by paying attention to what the authors call 'spatial bias'. This project forms part of a long-running debate among palaeobiologists, that was started by Raup's influential 1972 paper in *Science* in which he suggested that the 'fossil record' might be more bias than signal. His argument was that our knowledge decays backwards in time and so the apparent rise of palaeodiversity from the dim distant past to the present day could simply record those biases (geological and human). What if, he asked, global diversity had always been more or less constant, and we simply sample an increasing amount of it from the Cambrian to the present?

In their study, the current authors sampled all tetrapods, and removed marine forms (ichthyosaurs, plesiosaurs, whales etc) and flying forms (primarily pterosaurs, birds and birds). This is fine. The plots of raw data and sampling-standardised data in Figure 1c appear to show the classic somewhat exponential curves, with very low levels of diversity through the late Palaeozoic and Mesozoic, and a substantial uplift through the Late Cretaceous and Cenozoic to reach levels 3–4 times Late Cretaceous levels in the Neogene/ Quaternary.

The key contribution here is to apply corrections for 'spatial bias', an idea posited before by Close et al. (2017). This controls for variation in the palaeogeographic extent of fossil record sampling within intervals. Basically, their estimate of habitable area of the land surface (green curve in Fig. 1a) tracks the curves for palaeodiversity (Fig. 1c, both curves), and if one accepts that the amount of habitable area drives recorded diversity, then it makes sense to correct the one by the other and achieve a more or less flat line. Hence, they conclude that diversity of tetrapods has not risen exponentially through time, or at least through the last 100 Myr, and that global diversity was constrained over spans of tens or hundreds of millions of years, for all tetrapods, and for subsets of tetrapods.

So, if their correction method based on habitable areas is valid, they are right; if not, their suggestion contributes nothing.

Figure 1A shows that habitable area increases by maybe 33% (20% to 30% of Earth's surface), whereas their metric of spatial sampling increases from 0 to 100% along a curve that is very close to that of uncorrected global species richness (Fig. 1B). Such close tracking suggests that the two signals are linked, but which drives which? It could be one of two explanations:

1. Spatial sampling drives apparent palaeodiversity, and so the latter should be corrected by the former, as it is a sampling signal.
2. Spatial sampling reflects actual increases in palaeodiversity, and so is a redundant signal with raw diversity.

As an initial observation, these close correlations used to be accepted as sure evidence that, for example, numbers of specimens, collections or localities provided a valid proxy for sampling, but their close tracking of the raw palaeodiversity signal on the other hand really reflected redundancy, as I argued many times, and as these authors now seem to accept.

Now, I know that spatial sampling is a real thing, and we demonstrated it in our study of the British fossil record (Dunhill et al. 2014), but I hope the authors will accept that everything they say is not self-evident, and it is worth taking a critical look at what they are doing. Can they categorically rule out that for some or all clades, there might have been times when real distributions were patchy or limited to certain continental areas? In other words, since the Carboniferous, the ways in which the surface of the Earth was habitable have varied. Three things:

1. In the Carboniferous, perhaps tetrapods occupied only 10% of the apparently 'habitable' land surface because most were tied to freshwaters (temnospondyls, anthracosaurs, lepospondyls), and the areas covered by plants and soils perhaps extended only a few 100 m away from water courses, leaving hills, mountains and centres of continents as bare rock. Very likely continental centres remained barren through the Permian and Triassic as well – that could be >50% of the Earth devoid of tetrapod life in reality. Therefore, correcting the raw diversity values (whether for continents or the world) by assuming those areas ought to have been habitable according to modern standards would artificially inflate the total diversity estimates, perhaps by 100% or more.
2. Pangaea split in stages and the degree of endemism of terrestrial tetrapods has perhaps changed through time. Some might say this has been a big influence on diversification at various levels; others might say it has not been. But, again, applying a method that takes account only of total land area of particular regions or the world might miss any impact of the combined continental position/ sea level on endemism and hence total global diversity.
3. Perhaps climatic changes coupled with changing physiologies would have other non-uniform effects. As you generate climate belts, you generate diversity of habitats (maybe an effect of doubling or trebling the diversity of habitat types from say Jurassic to Neogene), and this might have different effects on endotherms and ectotherms and their ability to generate diversity. Is this factored into the spatial metrics?

I know this is repeating stuff from Sahney et al. (2010) and elsewhere, but in the context of application of a single method to all data through a massively changing world might give some pause for thought.

Two further thoughts:

1. By excluding birds, you remove 10,000 modern species, a fair chunk of modern tetrapod biodiversity, but I can follow the reasoning in terms of sampling.
2. More fundamentally though is that the study makes its very large claims based only on fossil data, in other words clipping the raw global diversity at the modern end of the time scale to maybe 1100 species, instead of the actual 30,000 species (or 20,000 species without birds). This

means, I assume, what you are saying is that 'corrected' global diversity for the Cenozoic is perhaps 20,000 species throughout, and half that, 10,000 species, throughout the whole of the Mesozoic.

Line 47: I'm not sure these studies were based on 30-year-old compilations. We used the Fossil Record 2 (1993) in the 1996 paper, which is now 26 years old, but was only 3 years old then (ref. 4). For Sahney et al. (2010, ref. 8) these data were then 17 years old.

Lines 265-266: 'Our results are consistent with a growing body of evidence [5,9,10,15,16,28,33,34] for constrained diversification within the terrestrial realm.' Interesting – I suspect a lot of terrestrial ecologists and phylogenomics people will be a bit startled. Maybe you could explain what these constraints are that mean changing geographies through the Cenozoic have had negligible effect on summed global diversity. Also, perhaps a word about the apparent explosion in diversity of clades like murid rodents (or passeriform birds – but birds are excluded) – they were presumably then replacing a precursor equally diverse clade that disappeared or shrank? There can be no suggestion in a world of constraint such as you propose that such clades could actually have hit on a smart new mode of life and that they really did diversify explosively and without displacing pre-existing species.

And maybe a word to those who are interested in gymnosperms and angiosperms – in the modern world, angiosperm-dominated forests tend to be rich in life of all sorts, whereas gymnosperm-dominated forests are not. In terms of animal diversity this can be an effect of one or more orders of magnitude. Perhaps the switch from one to the other through the Cretaceous did nothing to insects, lizards, mammals, etc? I certainly can't prove the shift in dominant plants did drive diversity up, but many would accept that likely this did, and it might be good if you could explain why in fact the Cretaceous angiosperm expansion made no difference to global diversity of tetrapods.

Mike Benton

Referee: 2

Comments to the Author(s)

This paper concerns the pattern of tetrapod species richness across the last c. 300 million years, and more particularly whether it fits an expansionist or constrained model of diversification. The methods used are appropriate and cutting edge and further emphasise the importance of accounting for spatial factors in palaeontological estimates of taxonomic richness. Importantly, the results strongly favour the constrained hypothesis, aka, "diversity dependence" or a "carrying capacity" model, with an upwards shift in this value coincident with the K-Pg boundary.

I am happy to support publication almost as is, except for a few minor corrections below. Aside from this, I personally think the manuscript could be improved by considering what we might expect the pattern of tetrapod diversity to be over the sampling window. For example, major innovations we might expect to increase richness are: 1) the origin of tetrapod herbivory, 2) the origin of flowering plants and, 3) the expansion of grasslands. None of these are coincident with the one shift the authors favour (at the K-Pg). Additionally, two major geographic considerations seem undiscussed. Firstly, the effects of the assembly and particularly the breakup of the Pangaean supercontinent on richness has been much discussed in the literature (e.g., Jordan et al. 2016, *Phil. Trans. Roy. Soc. B.*, 371, 20150221; Vavrek 2016, *Biol. Lett.*, 12, 20160528). Both of those papers seemingly make predictions that are not supported by the results presented here and should perhaps be used as additional context for interpretation. Secondly, the authors' results appear to rule out a species-area effect in tetrapod biogeography, something that studies of extant richness typically support. This seems like an additional and major implication of the results that the authors do not currently discuss.

Line-by-line comments:

L82 - "All data downloads were performed in the analysis R scripts" Grammatically this made no sense to me. Suggest rewording.

L83 - "were recorded as text files deposited on Dryad" Missing "and"?

L88 - "comprised 10,699 collections (= localities)" I am not sure this is a safe assumption (i.e., collections do not always perfectly represent localities). However, the authors have applied spatial grouping methods that account for this issue, so perhaps this could be stated instead.

L91 - "if over 50% of their temporal duration was contained" Somewhat pedantic, but do tetrapod occurrences truly have a duration? I would have thought this really means temporal error. A single fossil has no duration at all. Suggest rewording to clarify. A deeper question might be what this scheme means for temporal patterns. I.e., does it "clump" occurrences, "smooth" them etc. What about very poorly temporally constrained occurrences? And perhaps ultimately, what proportion of the data does this represent?

L112-115 - The numbering here is incorrect (3 is repeated).

L118 - Equal-size, but what shape are these? A grid implies rectangles, but obviously the curvature of the surface of a sphere significantly complicates this. Additionally, one imagines membership of a grid cell is dependent on the "anchoring" of this grid as any covering of a sphere is essentially infinitely rotatable. What was the chosen origin? Or, most simply, can a figure showing this grid on a map be included in the SI.

L133 - "in order to excluding" Grammar.

L257 - Alternatively, perhaps mammals have been oversplit? Or this is an issue with temporal binning, time-averaging higher-temporal resolution collections of short duration taxa together? I do not think it is on the authors to solve this question, but this seems like an obvious "further work" part of the discussion.

L259 - "Palaeozoic" should be "Paleozoic". This is a formal name not a UK- vs US-spelling issue.

L384 - "500 km spacings", earlier (L118) 100 km spacings were stated and in the SI 200 km. Some clarity would help here.

L402 - Assuming these are Phylopic silhouettes the artists should be given specific credit and some may require permission to use. Please check the web site: <http://phylopic.org/about>.

Referee: 3

Comments to the Author(s)

Overall this is an excellent paper, the analyses are well thought out and executed and the manuscript is clearly written. I don't have methodological or inference concerns. I have some minor comments in the attached pdf on the writing in the methods section. I have no comments on the supplementary information. The online tool is a very useful addition that helps in understanding the analyses and patterns.

I have one specific thing to highlight from the comments in the pdf. The analyses in this paper are complex with many steps that become hard to keep track of, I think it would be very useful to have a methods figure with a schematic of the procedure to make it more easily understandable. A flow chart of the pipeline, something like in Gilbert and Escarguel 2017 "Evaluating the accuracy of biodiversity changes through geologic times: from simulation to solution". Or even better explanatory images of toy examples to go with it. I think it's worth taking the time to explain things in straightforward terms to make it as readable (and therefore as citable) as possible given the potentially broad audience that might be interested in this.

One other question is which figure should be used for the main manuscript figure. Perhaps the authors feel it would be misleading to select one of the 12 boxes and present that (if so ok leave things as they are, or perhaps even just switch Fig 2 and fig S3), but I think if it is made clear in the caption that all the other approaches show broadly the same result I think that is ok. The paper presents a convincing 'global' terrestrial diversity plot for the Phanerozoic, I know its not

possible to actually do a global curve because of the nature of the data and analyses but at the moment I think having 12 virtually identical plots in the main results figures buries the message and impact of the result. From my perspective the ideal way to make very clear the main result would be to select one of the plots from supplementary figure 3 (maybe squares 3000 or sqs 3000) that has the model lines on it as the main results figure then put the rest in the supplement, or have the current fig 2 as a fig 3.

Author's Response to Decision Letter for (RSPB-2019-2254.R0)

See Appendix A.

RSPB-2020-0372.R0

Review form: Reviewer 1 (Michael J. Benton)

Recommendation

Accept as is

Scientific importance: Is the manuscript an original and important contribution to its field?

Good

General interest: Is the paper of sufficient general interest?

Good

Quality of the paper: Is the overall quality of the paper suitable?

Good

Is the length of the paper justified?

Yes

Should the paper be seen by a specialist statistical reviewer?

No

Do you have any concerns about statistical analyses in this paper? If so, please specify them explicitly in your report.

No

It is a condition of publication that authors make their supporting data, code and materials available - either as supplementary material or hosted in an external repository. Please rate, if applicable, the supporting data on the following criteria.

Is it accessible?

Yes

Is it clear?

Yes

Is it adequate?

Yes

Do you have any ethical concerns with this paper?

No

Comments to the Author

I disagree with the conclusions of the paper, because I don't think the method does what is claimed. The geographic occupancy curve (green, Fig. 1B) and the face-value diversity curve (red, Fig. 1A) could be the same because they both to some extent track the same signal, expanding occupancy of the globe, rather than simply a metric of improving geographic coverage of fossil finds. However, the method deserves wider consideration, and it is well applied and the current paper explains it more clearly than before, so I am happy to provide good ratings for the paper.

Decision letter (RSPB-2020-0372.R0)

12-Mar-2020

Dear Dr Butler

I am pleased to inform you that your Review manuscript RSPB-2020-0372 entitled "The apparent exponential radiation of Phanerozoic land vertebrates reflects spatial sampling biases" has been accepted for publication in Proceedings B. Congratulations!!

The referee(s) do not recommend any further changes. Therefore, please proof-read your manuscript carefully and upload your final files for publication. Because the schedule for publication is very tight, it is a condition of publication that you submit the revised version of your manuscript within 7 days. If you do not think you will be able to meet this date please let me know immediately.

To upload your manuscript, log into <http://mc.manuscriptcentral.com/prsb> and enter your Author Centre, where you will find your manuscript title listed under "Manuscripts with Decisions." Under "Actions," click on "Create a Revision." Your manuscript number has been appended to denote a revision.

You will be unable to make your revisions on the originally submitted version of the manuscript. Instead, upload a new version through your Author Centre.

- 1) A text file of the manuscript (doc, txt, rtf or tex), including the references, tables (including captions) and figure captions. Please remove any tracked changes from the text before submission. PDF files are not an accepted format for the "Main Document".
- 2) A separate electronic file of each figure (tiff, EPS or print-quality PDF preferred). The format should be produced directly from original creation package, or original software format. Please note that PowerPoint files are not accepted.
- 3) Electronic supplementary material: this should be contained in a separate file from the main text and the file name should contain the author's name and journal name, e.g. `authorname_procb_ESM_figures.pdf`

All supplementary materials accompanying an accepted article will be treated as in their final form. They will be published alongside the paper on the journal website and posted on the online figshare repository. Files on figshare will be made available approximately one week before the

accompanying article so that the supplementary material can be attributed a unique DOI. Please see: <https://royalsociety.org/journals/authors/author-guidelines/>

4) Data-Sharing and data citation

It is a condition of publication that data supporting your paper are made available. Data should be made available either in the electronic supplementary material or through an appropriate repository. Details of how to access data should be included in your paper. Please see <https://royalsociety.org/journals/ethics-policies/data-sharing-mining/> for more details.

<http://datadryad.org/submit?journalID=RSPB&manu=RSPB-2020-0372> which will take you to your unique entry in the Dryad repository.

Once again, thank you for submitting your manuscript to Proceedings B and I look forward to receiving your final version. If you have any questions at all, please do not hesitate to get in touch.

Sincerely,

Dr John Hutchinson, Editor

Associate Editor

Board Member

Comments to Author:

Thank you to authors for producing an exhaustive response to the initial round of reviews provided by the three expert referees and myself. The second submission has been read by Reviewer 1 and myself. Reviewer 1 still does not completely agree with some of the interpretations/conclusions of the study but nevertheless feels that work (in its current form) deserves publication to stimulate discussion and future work. I am very happy with the revisions made in response to reviewer 3's and my own concerns about accessibility of the MS to the general reader/non-expert. The methods, and particularly why certain steps in the methods are there, are much clearer to me conceptually after reading this version. Similarly, I think the discussion section is much improved by wider treatment of some the issues raised in the first round of reviews. I'm happy to recommend publication of this work in Proceedings B.

Reviewer(s)' Comments to Author:

Referee: 1

Comments to the Author(s).

I disagree with the conclusions of the paper, because I don't think the method does what is claimed. The geographic occupancy curve (green, Fig. 1B) and the face-value diversity curve (red, Fig. 1A) could be the same because they both to some extent track the same signal, expanding occupancy of the globe, rather than simply a metric of improving geographic coverage of fossil finds. However, the method deserves wider consideration, and it is well applied and the current paper explains it more clearly than before, so I am happy to provide good ratings for the paper.

Sincerely,

Proceedings B

Author's Response to Decision Letter for (RSPB-2020-0372.R0)

See Appendix B.

Decision letter (RSPB-2020-0372.R1)

16-Mar-2020

Dear Dr Butler

I am pleased to inform you that your manuscript entitled "The apparent exponential radiation of Phanerozoic land vertebrates reflects spatial sampling biases" has been accepted for publication in Proceedings B.

Open Access

Paper charges

Sincerely,
Editor, Proceedings B
<mailto:proceedingsb@royalsociety.org>

Appendix A

Dear Dr Hutchinson,

Thank you for considering our manuscript for publication in *Proceedings of the Royal Society B*. The three referees and Associate Editor provided insightful and constructive comments that have enabled us to improve the manuscript.

We are especially pleased that two of the three reviewers (Referees 2 and 3) supported publication of our study almost as-is, barring some minor corrections. Referee 1 voiced more substantial concerns, but most of these appear to stem from misunderstandings of our methods, which we have addressed in our point-by-point responses (below), as well as through striving to increase the clarity of our presentation in the manuscript. The Associate Editor echoed some of the concerns voiced of referees and raised a couple of separate issues, which we have also been able to address in our point-by-point responses, and through minor revisions to the manuscript and figures.

The revised manuscript contains some minor changes. Most notably, we have:

1. Edited the text throughout to make it clearer that our work analyses regional, not global, diversity.
2. Substantially revised the entire manuscript to increase clarity, with a particular emphasis on the Methods section.
3. Added a paragraph to the Discussion which considers some events in Phanerozoic terrestrial tetrapod evolution (raised by reviewers) that have been hypothesised to have catalysed increases in diversity, yet which do not seem to have influenced the patterns we recover.
4. Moved some important figures from the supplement to the main text.
5. Added a new figure explaining the method we use for estimating regional-scale (i.e., spatially-standardised) diversity.
6. Added supplementary figures (S1–S4) showing examples of the grid-cell binning scheme used, along with other palaeomaps visualising the steps involved in our spatial standardisation procedure.
7. Added a statistical test to quantify the relative explanatory roles of worldwide spatial sampling, continental area and non-marine sediment extent on “global” diversity (a generalised least-squares model using a first-order autoregressive correlation structure; Table 2).
8. Changed the main bivariate comparisons of global time series (e.g. between diversity, spatial sampling and continental area) to use first-differenced variables, in order to guard against spurious time series correlations, and added additional panels to Fig. 1 showing relationships between changes in diversity and changes in continental area.
9. Added supplementary figures showing bivariate relationships between time series of “global” diversity, the palaeogeographic spread of the fossil record, continental area and non-marine sediment extent in more detail (as raw relationships, first-differences and detrended using ARIMA models; Figs S5–S8).
10. Simplified the main-text figure showing regional-scale diversity patterns for non-flying terrestrial tetrapods by restricting it only one spatial scale, as suggested by Reviewer 3 (Fig. 4). Additional spatial scales are still shown in Fig. S9.
11. Moved diversity patterns computed using the grid-cell rarefaction (GCR) procedure to a supplementary figure (Fig. S10). SQS was the only richness estimator to be computed using GCR, due to heavy computational demands, and we decided that it was fairer to compare all richness estimators in the main figures on an equal footing (i.e., without GCR).

We thank you again for overseeing this valuable process, and hope that the changes we have made in our revised manuscript have improved it considerably.

Sincerely, and on behalf of the authors,

Roger Close

Summary of Authors' Response

Referee 1 listed a set of concerns about our method for estimating diversity from standardised spatial regions. However, these concerns seem to originate from misunderstandings of our analytical procedure and the spatial scale that our study focusses on (regional, not global). We believe that we have been able to adequately address these in our point-by-point responses (below), and by editing the manuscript to try and improve clarity (including adding extra figures to the supplement to explain the methodology).

Referees 1 and 2, along with the Associate Editor, queried why our results do not appear to support major roles for certain events that have been hypothesised to have catalysed increases in tetrapod diversity (such as the origins of tetrapod herbivory, flowering plants, the radiations of groups that are speciose today such as murid rodents, and continental fragmentation). Referee 1 also suggests that these events seem incompatible with a model of constrained diversity. We address these points in more detail in our point-by-point responses (below), and in a paragraph added to the Discussion.

Referee 3 and the Associate Editor were concerned that the analyses in our study are complex and hard to keep track of. To address this, we have added a bullet-point summary of the analytical procedure to the Methods, along with a supplementary figure showing key steps in our spatial standardisation procedure. We have also strived to improve the clarity of the main text wherever possible, especially in the Methods.

The Associate Editor also queried the ability of our large fossil dataset to answer the questions we pose about patterns of regional-scale diversity in Phanerozoic terrestrial tetrapods. While we understand these concerns, we believe them to be unfounded, and have addressed them in our point-by-point responses (below).

Lastly, all three referees and the Associate Editor also raise a number of more minor points, which we have addressed in our point-by-point responses and in our revised manuscript. We have also caught and amended some errors, such as counts of occurrences, collections and unique species that dated from an earlier version of our analysis.

Point-by-point responses to reviewers

Associate editor

Comments to Author:

Thank you for the opportunity to review this very interesting paper. Herein the authors re-examine species-level diversity distributions in (most) terrestrial vertebrates through the last ~400 million years, taking into account spatial sampling bias for the first time in this context. They use this approach to examine the relative support for two competing models of diversity through geological time given the fossil data available: diversity expanded through time versus diversity remained more or less constant through time. The implication of the latter model is that the observed expansionist trends in raw data through time are in fact driven by spatial sampling bias in the fossil record, and thus not a true evolutionary signal. This is a very topical area and one likely to be of interest to a broad range of biological and geological scientists. All three reviewers are positive about the goals and approaches of the study, but have forwarded a number of criticisms and/or suggested improvements that should be made before publication. I find myself agreeing with all these suggestions, and perhaps even going a little further in some cases.

As a biologist working outside this immediate area of investigation, I found it extremely difficult to follow (even conceptually at times) the many, many steps in data processing and analysis that ultimately lead to spatially considered/corrected diversity analyses. This issue is raised by expert Reviewer 3 and I agree that at present there is little hope that the general reader of Proceedings B will be able to judge exactly what is being

done, why it is being done and how valid or crucial certain assumptions are to the big picture. Reviewer 3 highlights some examples in their annotated pdf, and I would also highlight sentences like “Occurrences were assigned to a bin if over 50% of their temporal duration was contained within a single bin.” Adding explanations of why these kinds of steps were done, and in other cases some evaluation of the assumptions underpinning such methodological decisions, would be very helpful to the readership. ...

Thank you for taking the time to read our manuscript carefully, and to provide your own thoughtful feedback in addition to that of the reviewers.

We appreciate these criticisms about the clarity of the methods we used in our analysis, and have strived to make them easier to understand in our revised submission. To this end, we comprehensively revised the Methods section, which now starts with a bullet-point summary of key steps. We experimented with conveying this information in a flow chart as Reviewer 3 suggested, but decided that this was less effective than a simple bullet-point summary. However, we have also added a supplementary figure illustrating key steps in the spatial standardisation procedure we use to obtain estimates of regional-scale diversity. Where necessary, we have also added additional explanations for particular methodological steps. Many of the procedures we use, such as the binning scheme, are relatively standard practices in palaeobiology studies; although we have added clarifications to these, we have done so bearing in mind the space constraints of *Proceedings B*.

... I would also like some critical thought or evaluation of raw data being used to quantify land area/spatial distributions in the fossil record. Surely these data are estimations and contain error and potentially even some systematic temporal bias in that error. Or not?

We use the spatial distribution of fossil localities with well-defined palaeocoordinates to quantify the palaeogeographic extent of the known fossil record for each interval. The strength of the correlation between geographic spread and estimated richness is very great, and is unlikely to be the result of errors. Minor errors would primarily arise from recording modern-day geographic coordinates inaccurately in the Paleobiology Database, and from tectonic rotations used to recover palaeocoordinates. However, for most of the standardised palaeogeographic regions that we analyse (i.e., subsamples of fossil localities with approximately equal geographic extents), the localities come from regions of the globe that are linked on a single tectonic plate that moves as a rigid unit. Therefore, the error associated with these estimates are, for our purposes, negligible. This caveat has been added to the supplementary methods section.

Reviewers 1 and 2 ask for a much broader and perhaps more balanced discussion of the results and have independently cited similar topics that warrant detailed treatment given the specific trends and overall qualitative interpretation favoured by the authors.

We have addressed these issues raised by Referees 1 and 2 in our point-by-point responses to their reviews (below), and have added a paragraph to the discussion of our manuscript which discusses these issues.

Many of these things occurred to me when first reading the study, and in fact certain (almost throwaway) statements in the paper made me think even more critically than Reviewers 1 and 2 have about the take home message carried by the paper. The paper is set up to evaluate the validity of the expansionist model of diversity. Having put forward evidence that the expansionist model is not supported, the authors favour a model of relative stasis in diversity through time. The authors are mostly (but perhaps not always) careful not to push the support for stasis model too hard, but ultimately I was left feeling that a third equally valid interpretation was not discussed at all in the paper (at least not explicitly): that the raw fossil (and land area?) data is fundamentally not good enough to pick up the true diversity signal, whatever it is. There's little if any explicit discussion of this in the paper, but surely it is fundamental to deciding between the competing models. I'm not suggesting that the authors somehow quantify exactly how incomplete the Phanerozoic fossil record actually is, but there are number of their own points that, to me, make the conclusion that the raw fossil data isn't good enough to discern between competing models inescapable, at least as a possible conclusion. These include, for example, the significant impact that well-sampled mammal assemblages have on particular time bins, the fact that relative mammalian vs squamate diversity is reversed at certain points in recent history, and

not least the statement in the methods that “Estimating truly representative ‘global’ diversity curves for terrestrial tetrapods is therefore not possible based on our current knowledge of the fossil record.” It seems possible that the additional discussion asked for by Reviewers 1 and 2 may lead to situation where “the data ultimately isn’t good enough for us to tell” feels like an even more reasonable conclusion. Maybe I’m being overly pessimistic, or missing something fundamental, but I feel at the very least this conclusion should be explicitly discussed (and if valid to do so, objectively discounted) in the paper.

There is a long history of estimating richness from fossil data, and this has been influential on thinking about diversification. The work we presented is probably the most circumspect attempt at characterising underlying richness variation through the Phanerozoic in the terrestrial realm so far. We are naturally quite concerned about the usefulness of fossil occurrences to inform questions of diversity, and this is evident from our approach: for example, why the irregularities of the data are clearly flagged in the manuscript.

Our analysis is restricted to spatial regions of comparable geographic size primarily to ensure that all data points meet fairly stringent sampling criteria, and thus allow fair comparisons. This is not possible at the global level—or even the level of entire continental regions, which themselves tend to be subject to incomplete spatial coverage through time. We have tried to clarify this as much as possible at various points throughout the manuscript. For example, we have now included a section towards the end of the introduction: “Here, we present the first regional-scale diversity patterns for terrestrial tetrapods that cover their entire Phanerozoic evolutionary history, while adequately correcting for key biases. In doing so, we interpret the structure of the fossil record as an array of well-sampled palaeogeographic regions that contain useful information about regional palaeodiversity, but which are only indirectly informative about true global palaeodiversity.”

Most of the issues and suggested shortcomings of our work that were raised by Referee 1 only apply to global-scale analyses, which are simply beyond the scope of the fossil record at present (and likely forever, because the extent of non-marine sediments decays exponentially further back in time [1]).

Of course, no-one so far has succeeded at removing all biases in this type of study. However, sampling biases in the fossil record pretty much all act in favour of finding increases in diversity towards the present, and therefore make it less likely that we would recover a pattern suggesting relative stasis.

In any case, we hope that the Associate Editor agrees that the main take-home message of our paper—that the long-favoured near-exponential increases in terrestrial tetrapod diversity are an artefact of increasing spatial sampling towards the present—is a robust and important finding that deserves widespread attention.

Given all this, it is my recommendation that the authors be encouraged to submit a revised version of the manuscript to Proceedings B for further consideration.

Referee: 1

Comments to the Author(s)

This contribution does two novel things. First, it provides a set of palaeodiversity curves for terrestrial tetrapods based on current data, as downloaded from the Paleobiology Database, and (2) it ‘corrects’ these data by paying attention to what the authors call ‘spatial bias’. This project forms part of a long-running debate among palaeobiologists, that was started by Raup’s influential 1972 paper in *Science* in which he suggested that the ‘fossil record’ might be more bias than signal. His argument was that our knowledge decays backwards in time and so the apparent rise of palaeodiversity from the dim distant past to the present day could simply record those biases (geological and human). What if, he asked, global diversity had always been more or less constant, and we simply sample an increasing amount of it from the Cambrian to the present?

We think that there is a key difference between past studies, such as the seminal paper by Raup (1972), and our present study. Whereas the debate about bias had previously been framed in fairly abstract terms, such as rock amount, we explicitly focus on bias resulting from the spatial extent and distribution of fossil localities. This directly controls estimated diversity through species-area effects, which are ubiquitous. Because the area sampled by the fossil record – and the extent of the non-marine sedimentary record itself – increase exponentially through the Phanerozoic, so-called ‘global’ diversity curves have little meaning. ‘Redundancy’ cannot explain the correlation between diversity and spatial sampling, because changes in spatial sampling do not reflect changes in actual global terrestrial area. The only viable solution for estimating patterns of diversity through the Phanerozoic is to identify sets of fossil localities with comparable spatial extents—i.e., directly comparable palaeogeographic regions (this is what Referee 1 refers to as our ‘correction’ of the data).

In their study, the current authors sampled all tetrapods, and removed marine forms (ichthyosaurs, plesiosaurs, whales etc) and flying forms (primarily pterosaurs, birds and birds). This is fine. The plots of raw data and sampling-standardised data in Figure 1c appear to show the classic somewhat exponential curves, with very low levels of diversity through the late Palaeozoic and Mesozoic, and a substantial uplift through the Late Cretaceous and Cenozoic to reach levels 3–4 times Late Cretaceous levels in the Neogene/ Quaternary.

The key contribution here is to apply corrections for ‘spatial bias’, an idea posited before by Close et al. (2017). This controls for variation in the palaeogeographic extent of fossil record sampling within intervals. Basically, their estimate of habitable area of the land surface (green curve in Fig. 1a) tracks the curves for palaeodiversity (Fig. 1c, both curves), and if one accepts that the amount of habitable area drives recorded diversity, then it makes sense to correct the one by the other and achieve a more or less flat line. Hence, they conclude that diversity of tetrapods has not risen exponentially through time, or at least through the last 100 Myr, and that global diversity was constrained over spans of tens or hundreds of millions of years, for all tetrapods, and for subsets of tetrapods.

So, if their correction method based on habitable areas is valid, they are right; if not, their suggestion contributes nothing.

Crucially, Referee 1 appears to have misunderstood our analytical approach for controlling for spatial bias. The reviewer states that we “correct the one [i.e., global diversity estimates] by the other [habitable terrestrial area] and achieve a more or less flat line”. This comment suggests that we corrected curves of global diversity using estimates of habitable area, via some kind of residuals-based or upscaling method. This is not the case. We do not attempt to estimate global diversity (see below): rather, we directly estimate regional-scale diversity.

Our method is, at heart, very straightforward: we simply estimate diversity for palaeogeographic regions of approximately equal size. We draw sets of fossil localities that have approximately equal spatial extents, and use these to define palaeogeographic regions for which we directly estimate regional diversity. This allows us to standardise the extent to which truly inhabited areas have actually been sampled.

Because of the inconsistent and patchy nature of spatial sampling in the Phanerozoic fossil record, this is only way to ensure fair comparisons of diversity (other than analysing smaller spatial scales, such as local community richness [2]). Spatial sampling in the fossil record varies dramatically through intervals of geological time: regions sampled by fossil localities vary substantially in size, number and identity/location through the Phanerozoic—and, crucially, overall sampling increases exponentially towards the present. If we don’t acknowledge and account for this variability, then it is simply not possible to make fair comparisons of diversity levels through time.

An analogy may help clarify the problem: suppose we wanted to estimate tetrapod diversity on Earth today. We would get very different answers if our ‘global’ diversity estimate was based on sampling only North America, or only North America and Europe, or if we comprehensively sampled all continents. Because species-area relationships and global faunal provinciality are ubiquitous, we would spuriously estimate dramatically different diversity global levels. This is essentially what face-value global diversity curves of Phanerozoic tetrapods do, and this is why they are fundamentally flawed.

Additionally, Referee 1 seems to have misinterpreted the identities of the time series in Fig. 1, and this misunderstanding may also underpin some of his concerns about the paper. As indicated in the figure caption, habitable area is actually shown by the purple line, whereas spatial sampling (i.e., how that habitable area has actually been sampled in the fossil record) is shown by the green curve.

As is clear, face-value (observed) and ‘global’ sample-standardised species richness (Fig. 1C) do *not* track actual habitable area (Fig. 1A, purple curve), but *do* closely track spatial sampling of that habitable area (Fig. 1A, green curve). We have now made this clearer at various points in the manuscript, and included statistical tests that demonstrate that actual ancient habitable area is not closely related either to fossil tetrapod diversity or to the geographic size of the fossil record itself. Furthermore, a GLS model of “global” diversity as a function of spatial sampling, continental area and non-marine sediment extent (using a first-order autoregressive correlation structure) demonstrates a strong, statistically significant explanatory role only for spatial sampling. We think this provides strong evidence of spatial bias.

Figure 1A shows that habitable area increases by maybe 33% (20% to 30% of Earth’s surface), whereas their metric of spatial sampling increases from 0 to 100% along a curve that is very close to that of uncorrected global species richness (Fig. 1B). Such close tracking suggests that the two signals are linked, but which drives which? It could be one of two explanations:

1. Spatial sampling drives apparent palaeodiversity, and so the latter should be corrected by the former, as it is a sampling signal.
2. Spatial sampling reflects actual increases in palaeodiversity, and so is a redundant signal with raw diversity.

In order to accept that diversity drives spatial sampling, or that it is redundant with true diversity throughout the Phanerozoic, we would also need to accept that the global distribution of tetrapods has been increasing exponentially over the same interval. This is not supported by the evidence. If anything, the global range of tetrapods was even higher in the ancient past. In the mid-Cretaceous, over 100 myr ago, tetrapods were distributed from pole to pole, with a much broader latitudinal range for many groups enabled by greenhouse conditions. This is inconsistent with the exponential pattern, either in face-value global taxon counts or spatial sampling in the fossil record.

Moreover, the spatial extent of terrestrial sediments themselves grow exponentially towards the present (see Peters and Husson [1]). It is impossible that terrestrial tetrapod diversity also somehow drives exponential increases in the spatial extent of terrestrial sediments – this would imply that biotic diversity somehow exerts a control on the global tectonic processes that lead to the burial, preservation and exposure of terrestrial sediments.

As an initial observation, these close correlations used to be accepted as sure evidence that, for example, numbers of specimens, collections or localities provided a valid proxy for sampling, but their close tracking of the raw palaeodiversity signal on the other hand really reflected redundancy, as I argued many times, and as these authors now seem to accept.

While we agree with the referee that counts of collections or formations may have some problems as measures of sampling effort, we do not necessarily agree with their arguments about redundancy. However, that is outside of the scope of this paper and represents a debate for another day.

Now, I know that spatial sampling is a real thing, and we demonstrated it in our study of the British fossil record (Dunhill et al. 2014), but I hope the authors will accept that everything they say is not self-evident, and it is worth taking a critical look at what they are doing. Can they categorically rule out that for some or all clades, there might have been times when real distributions were patchy or limited to certain continental areas?

In other words, since the Carboniferous, the ways in which the surface of the Earth was habitable have varied. Three things:

1. In the Carboniferous, perhaps tetrapods occupied only 10% of the apparently 'habitable' land surface because most were tied to freshwaters (temnospondyls, anthracosaurs, lepospondyls), and the areas covered by plants and soils perhaps extended only a few 100 m away from water courses, leaving hills, mountains and centres of continents as bare rock. Very likely continental centres remained barren through the Permian and Triassic as well – that could be >50% of the Earth devoid of tetrapod life in reality. Therefore, correcting the raw diversity values (whether for continents or the world) by assuming those areas ought to have been habitable according to modern standards would artificially inflate the total diversity estimates, perhaps by 100% or more. ...

Importantly, we did not “[correct] the raw diversity values (whether for continents or the world) by assuming those areas ought to have been habitable according to modern standards”. This comment suggests that we are applying some kind of residual diversity correction which, crucially, we are not. This is a misunderstanding of the methods we apply, which we responded to in more detail above. We accept that misunderstandings of our approach may reflect poor clarity on our part and have worked to improve clarity of the methods and spatial scale of analyses.

To identify palaeogeographic regions suitable for estimating diversity, we used all collections from which tetrapods had been found. It is quite plausible that Carboniferous tetrapods occupied only a small fraction of the habitable continental area. Therefore, both the true and sampled distribution of Carboniferous tetrapods/fossils would be very limited and patchy. In fact, our analysis did not identify any sets of fossil localities (i.e., palaeogeographic regions falling within a predetermined size range) for Carboniferous tetrapods that met our criteria. As a result, our diversity patterns contain no points for this interval. However, the distribution of tetrapods did not remain this restricted for very long, and our analyses were able to estimate diversity for some Permian and Triassic regions. If, as Referee 1 suggests, large areas of the continents were uninhabitable during the Permian–Triassic, this would not pose a problem for our spatial standardisation method. We do not try to estimate diversity for regions where no tetrapods or fossils exist—only for regions where tetrapod fossils have been found, and which were therefore habitable. It is perhaps worth reiterating that we do not subsequently upscale diversity to the areas of entire continents or the globe. We focus solely on regional-scale diversity, and our methods will not therefore artificially inflate diversity.

... 2. Pangaea split in stages and the degree of endemism of terrestrial tetrapods has perhaps changed through time. Some might say this has been a big influence on diversification at various levels; others might say it has not been. But, again, applying a method that takes account only of total land area of particular regions or the world might miss any impact of the combined continental position/ sea level on endemism and hence total global diversity. 3. Perhaps climatic changes coupled with changing physiologies would have other non-uniform effects. As you generate climate belts, you generate diversity of habitats (maybe an effect of doubling or trebling the diversity of habitat types from say Jurassic to Neogene), and this might have different effects on endotherms and ectotherms and their ability to generate diversity. Is this factored into the spatial metrics?

I know this is repeating stuff from Sahney et al. (2010) and elsewhere, but in the context of application of a single method to all data through a massively changing world might give some pause for thought.

The first and most important point to make in response to the issues raised above is that our study does not analyse global diversity, for reasons given earlier. Regional-scale (or local-scale) comparisons are the only viable route for directly estimating diversity patterns through the Phanerozoic. As such, continental breakup probably exerts only a limited influence on patterns of diversity at the spatial scales we examine.

We agree that continental breakup could drive increases in global faunal provinciality, which could in turn increase total global diversity. Estimates by Vavrek (2016) based on extrapolating species-area relationships suggest that the breakup might be capable of ultimately driving roughly two-fold increases through the Mesozoic, the main interval of Pangean breakup.

This is now explained in the Discussion: “Neither do our analyses of regional diversity rule out some increase in global richness due to continental fragmentation (although we have shown that global diversity cannot currently be estimated). Modelling of species-area relationships suggests that this effect could have approximately doubled global terrestrial tetrapod biodiversity between the Triassic and Late Cretaceous, during the main interval of Pangean fragmentation [43]. Pangean fragmentation was largely complete by the end of the Cretaceous, and it seems unlikely that the comparatively minor continental rearrangements that occurred during the Cenozoic could have driven the proposed ten-fold increase in global diversity recovered by influential previous work [5,9].”

It is particularly worth noting that the breakup of Pangea was largely complete by the end of the Cretaceous. In our discussion, we do acknowledge that there may have been increases in global diversity resulting from factors such as global faunal provinciality. But irrespective of this fact, direct estimates of true global diversity curves are not accessible from our current knowledge of the fossil record—and this is possibly forever unknowable, because the spatial extent of non-marine sediments decays exponentially into the past.

Two further thoughts:

1. By excluding birds, you remove 10,000 modern species, a fair chunk of modern tetrapod biodiversity, but I can follow the reasoning in terms of sampling.

As the reviewer notes and as discussed in the supplement there are very good reasons for excluding the bird record. However, in our recent study in *Nature Ecology & Evolution* [2], we showed that face-value estimates of avian local species richness (as quantified through rare, exceptional localities) barely increased at all between the Eocene and the Plio-Pleistocene. This is explained in the Discussion of the current work: “Although limitations of the fossil record prohibit us from analysing regional-scale flying tetrapod diversity here, within-community patterns suggest these groups (birds, bats, pterosaurs) were also subject to long-term constraints [21]. These patterns suggest that the early diversification of birds resulted in the stepwise addition of substantial species richness to terrestrial ecosystems [10], with limited subsequent increases [21] that mirror the patterns of tetrapod richness documented here.”

2. More fundamentally though is that the study makes its very large claims based only on fossil data, in other words clipping the raw global diversity at the modern end of the time scale to maybe 1100 species, instead of the actual 30,000 species (or 20,000 species without birds). This means, I assume, what you are saying is that ‘corrected’ global diversity for the Cenozoic is perhaps 20,000 species throughout, and half that, 10,000 species, throughout the whole of the Mesozoic.

Our focus is on regional-scale diversity, not global diversity. Sampling equivalently-sized regions in the present day would never result in richness estimates that even approach total global counts of tetrapods.

Furthermore, we disagree with the criticism that we only use fossil data for estimating Phanerozoic diversity patterns. The only alternatives are to use neontological occurrence data (e.g., using range-through methods) or phylogenetic trees of extant taxa, and both are severely compromised by pull-of-the-Recent biases that virtually guarantee finding large, artefactual increases in diversity close to the present.

Line 47: I’m not sure these studies were based on 30-year-old compilations. We used the Fossil Record 2 (1993) in the 1996 paper, which is now 26 years old, but was only 3 years old then (ref. 4). For Sahney et al. (2010, ref. 8) these data were then 17 years old.

We have changed this phrasing to “decades old”.

Lines 265-266: ‘Our results are consistent with a growing body of evidence [5,9,10,15,16,28,33,34] for constrained diversification within the terrestrial realm.’ Interesting – I suspect a lot of terrestrial ecologists and phylogenomics people will be a bit startled. Maybe you could explain what these constraints are that mean changing geographies through the Cenozoic have had negligible effect on summed global diversity. ...

The crucial point here is that we do not attempt to estimate summed global diversity for Phanerozoic terrestrial tetrapods, nor do we think that direct estimates of this parameter are even possible due to the limitations of the fossil record discussed above. We focus on regional-scale estimates, which—along with the local-community scale (examined for Phanerozoic terrestrial tetrapods by Close et al. [2])—corresponds to the spatial scales relevant to modern-day ecological studies that seek to understand the diversification process [3,4].

... Also, perhaps a word about the apparent explosion in diversity of clades like murid rodents (or passeriform birds – but birds are excluded) – they were presumably then replacing a precursor equally diverse clade that disappeared or shrank? There can be no suggestion in a world of constraint such as you propose that such clades could actually have hit on a smart new mode of life and that they really did diversify explosively and without displacing pre-existing species.

With respect to zero-sum replacements of clades: this is entirely possible. For example, rodents replaced multituberculates during the early Cenozoic. More generally, though, perhaps the expansion of these successful groups is part of reason for higher Cenozoic diversity. With respect to birds: they are excluded, but we noted above the patterns of local richness in birds between the Paleogene and Quaternary.

And maybe a word to those who are interested in gymnosperms and angiosperms – in the modern world, angiosperm-dominated forests tend to be rich in life of all sorts, whereas gymnosperm-dominated forests are not. In terms of animal diversity this can be an effect of one or more orders of magnitude. Perhaps the switch from one to the other through the Cretaceous did nothing to insects, lizards, mammals, etc? I certainly can't prove the shift in dominant plants did drive diversity up, but many would accept that likely this did, and it might be good if you could explain why in fact the Cretaceous angiosperm expansion made no difference to global diversity of tetrapods.

We have considered these proposed reasons for hypothetical diversity increases in the discussion of our revised manuscript (see also response to Referee 2, below). The hypothesised effects of the radiation of angiosperms on tetrapod diversity are not inconsistent with the patterns we present, because the success of flowering plants could be part of the reason for the large increase in diversity we observe following the K/Pg boundary. We would not necessarily expect an immediate impact on tetrapod diversity following angiosperm origins, because angiosperms remained relatively minor components of ecosystems for much of the Cretaceous. Other than that, however, the evidence we present in this study does not support a first-order role for these mechanisms in controlling tetrapod diversity at regional spatial scales.

Mike Benton

Referee: 2

Comments to the Author(s)

This paper concerns the pattern of tetrapod species richness across the last c. 300 million years, and more particularly whether it fits an expansionist or constrained model of diversification. The methods used are appropriate and cutting edge and further emphasise the importance of accounting for spatial factors in palaeontological estimates of taxonomic richness. Importantly, the results strongly favour the constrained hypothesis, aka, diversity dependence or a carrying capacity model, with an upwards shift in this value coincident with the K-Pg boundary.

I am happy to support publication almost as is, except for a few minor corrections below. ...

We thank Referee 2 for their very positive assessment of our study.

... Aside from this, I personally think the manuscript could be improved by considering what we might expect the pattern of tetrapod diversity to be over the sampling window. For example, major innovations we might expect to increase richness are: 1) the origin of tetrapod herbivory, 2) the origin of flowering plants and, 3) the expansion of grasslands. None of these are coincident with the one shift the authors favour (at the K-Pg). ...

Some of the issues that Referee 2 raises here are similar to those articulated by Referee 1 (see above). Specifically, however, our results do not shed light on diversity patterns associated with origin of tetrapod herbivory, because there are no spatial samples of fossil localities that meet our criteria for the temporal window in question (i.e., the Carboniferous). Our earliest spatially-standardised diversity estimates are from the Permian, after tetrapod herbivory had already appeared.

On the origin of flowering plants, we speculate that perhaps the *origin* itself is not important for diversity patterns, since angiosperms were rare during the early part of their evolution (e.g., in the Early Cretaceous) and only became more widespread and common in the latest Cretaceous. We might also speculate that angiosperms contributed to the explosion of diversity after K/Pg, since there is evidence that they became markedly more successful during this interval.

On the origins of grasslands, however, we would not necessarily expect this to catalyse substantial increases in diversity. Today, grasslands are not especially diverse relative to other (e.g., tropical) environments. The origin of grasslands might have been important for the diversity of some tetrapod clades, such as horses but, overall, the subtropical conditions that prevailed in North America during the Eocene are much more conducive to high diversity than temperate grassland environments. More generally, the onset of cooler climates in the mid-Cenozoic may partially explain the trend towards lower regional-scale diversity that we document. In any case, the appearance of grasslands does not appear to exert a first-order control on the patterns we observe.

... Additionally, two major geographic considerations seem undiscussed. Firstly, the effects of the assembly and particularly the breakup of the Pangaean supercontinent on richness has been much discussed in the literature (e.g., Jordan et al. 2016, *Phil. Trans. Roy. Soc. B.*, 371, 20150221; Vavrek 2016, *Biol. Lett.*, 12, 20160528). Both of those papers seemingly make predictions that are not supported by the results presented here and should perhaps be used as additional context for interpretation. ...

On the effects of the breakup of Pangea, please see our response to Referee 1 (above). To summarise, however, the key point is that our study estimates regional scale, not global, diversity. Direct estimates of global tetrapod diversity through the Phanerozoic are currently (and probably forever) unknowable, precisely because the record is so geographically incomplete—especially the further back in time we go. Indirect assessments of trends in true global diversity might be achieved through quantifying global faunal provinciality among sampled regions, while accounting for the incompleteness of global geographic sampling. The lead author is pursuing this in a separate study.

The Vavrek paper uses species-area relationships to predict that the breakup of Pangea would cause a doubling of *global* diversity. This is the predicted maximum effect of continental breakup after sufficient equilibration time (divergence/full turnover of faunas). This increase is small relative to the ten-fold increases over the last 100 myr previously proposed based on uncorrected ‘global’ diversity curves.

Furthermore, studies looking at whether the diversification process in the modern is dominated by diversity dependence or unconstrained diversification largely focus on local to continental scales. Constrained diversification under diversity dependence does not rule out the possibility of increases at the global or super-regional level resulting from greater geographic fragmentation/isolation and differentiation of faunas.

As stated in our response to Referee 1 and the Associate Editor, we have added a small paragraph to the Discussion in the main text, in which we mention some events in Phanerozoic tetrapod evolution that might be expected to increase diversity, but which apparently do not based on the results of our study.

... Secondly, the authors' results appear to rule out a species-area effect in tetrapod biogeography, something that studies of extant richness typically support. This seems like an additional and major implication of the results that the authors do not currently discuss.

We are afraid that we do not follow Referee 2's reasoning here. It is not true to say that our results rule out a species-area effect—either in the sense that larger areas support more species than smaller ones in total, or per unit area. The ubiquity of species-area relationships (SARs) in both the modern and in the fossil record is precisely what motivated us to apply the spatial standardisation methods used in our study.

Our study quantifies patterns of diversity at a range of standardised spatial scales, and it is true that results from the different scales we examine agree in overall pattern. This merely implies that the form of the species-area relationship (slope and intercept) does not vary dramatically through the Phanerozoic.

That said, species-area relationships do vary to some extent across geographic regions and through intervals of geological time. These variations in the form of the SAR explain some of the minor variations in relative diversity levels of spatially-standardised regions across spatial scales. The lead author is explicitly analysing variations in species-area relationships through time and space for a separate study, which will shortly be written up.

Line-by-line comments:

L82 - All data downloads were performed in the analysis R scripts Grammatically this made no sense to me. Suggest rewording.

This has been reworded – “All occurrence datasets were downloaded using the Paleobiology Database API [23], using function calls executed within the analysis R scripts (URLs used to perform these data downloads, together with all analysis scripts, are available on Dryad [XXX]).”

L83 - were recorded as text files deposited on Dryad Missing and?

Fixed.

L88 - comprised 10,699 collections (= localities) I am not sure this is a safe assumption (i.e., collections do not always perfectly represent localities). However, the authors have applied spatial grouping methods that account for this issue, so perhaps this could be stated instead.

Although collections can mostly be equated with localities, Referee 2 is right that collections sometimes represent slightly different groupings, such as different phases of exploration within the same locality. We have removed “(=localities)” and we explain what a collection is in more detail in the supplement.

Collections were only binned within 100 km equal-area grid cells for the spatial subsampling routines—collections are not spatially aggregated for estimating diversity (i.e., for computing occurrence-frequency distributions, etc.).

L91 - if over 50% of their temporal duration was contained Somewhat pedantic, but do tetrapod occurrences truly have a duration? I would have thought this really means temporal error. A single fossil has no duration at all. Suggest rewording to clarify. A deeper question might be what this scheme means for temporal patterns. I.e., does it clump occurrences, smooth them etc. What about very poorly temporally constrained occurrences? And perhaps ultimately, what proportion of the data does this represent?

Of course, Referee 2 is correct that individual fossils themselves do not have a duration as such (although an occurrence record might represent many fossils spanning a longer period of time).

To be consistent with the terminology used by the Paleobiology Database API, we have reworded this sentence to “Occurrences were assigned to a bin if that bin contained over 50% of the geologic time range associated with that occurrence (defined by the early and late bounds recorded by the ‘min_ma’ and ‘max_ma’ fields in the Paleobiology Database, in Ma).”

There’s no perfect approach to binning fossil occurrences—we have taken a pragmatic approach to maximising available sampling. Occurrences with extremely long durations are automatically removed by our binning approach, and we also removed such occurrences by excluding those for which stratigraphic scale was recorded as group in the Paleobiology Database.

We have added a note in the supplement recording the percentage of occurrences that were excluded by these criteria (i.e., binning scheme and stratigraphic scale).

L112-115 - The numbering here is incorrect (3 is repeated).

Fixed.

L118 - Equal-size, but what shape are these? A grid implies rectangles, but obviously the curvature of the surface of a sphere significantly complicates this. Additionally, one imagines membership of a grid cell is dependent on the anchoring of this grid as any covering of a sphere is essentially infinitely rotatable. What was the chosen origin? Or, most simply, can a figure showing this grid on a map be included in the SI.

The grid cells were generated using the R package *dggridR*, which uses a combination of hexagonal and pentagonal cells (needed to make them fit together on a sphere better, of course). We have clarified this in the text, and have also added a figure to the supplement showing how this grid-cell scheme looks.

L133 - in order to excluding Grammar.

Fixed.

L257 - Alternatively, perhaps mammals have been oversplit? Or this is an issue with temporal binning, time-averaging higher-temporal resolution collections of short duration taxa together? I do not think it is on the authors to solve this question, but this seems like an obvious further work part of the discussion.

It is possible that mammals are oversplit relative to other groups. We discussed these issues in the supplement for our recent paper on terrestrial tetrapod local richness in *Nature Ecology & Evolution* (Close et al. 2019), so we did not think it necessary to repeat these arguments here.

L259 - Palaeozoic should be Paleozoic. This is a formal name not a UK- vs US-spelling issue.

Fixed.

L384 - 500 km spacings, earlier (L118) 100 km spacings were stated and in the SI 200 km. Some clarity would help here.

This is not a typo; we simply used different grid-cell spacings for different purposes.

L402 - Assuming these are Phylopic silhouettes the artists should be given specific credit and some may require permission to use. Please check the web site: <http://phylopic.org/about>.

Fixed.

Referee: 3

Comments to the Author(s)

Overall this is an excellent paper, the analyses are well thought out and executed and the manuscript is clearly written. I don't have methodological or inference concerns. I have some minor comments in the attached pdf on the writing in the methods section. I have no comments on the supplementary information. The online tool is a very useful addition that helps in understanding the analyses and patterns.

We thank Referee 3 for their very positive assessment of our study.

I have one specific thing to highlight from the comments in the pdf. The analyses in this paper are complex with many steps that become hard to keep track of, I think it would be very useful to have a methods figure with a schematic of the procedure to make it more easily understandable. A flow chart of the pipeline, something like in Gilbert and Escarguel 2017 Evaluating the accuracy of biodiversity changes through geologic times: from simulation to solution. Or even better explanatory images of toy examples to go with it. I think it's worth taking the time to explain things in straightforward terms to make it as readable (and therefore as citable) as possible given the potentially broad audience that might be interested in this.

We thank the referee for this suggestion. We experimented with using a flowchart to describe the steps in our analysis procedure, but found this medium to be ineffective. Instead, our revised submission includes a bullet-point summary at the start of the Methods to highlight the key steps. We have also included a supplementary figure that illustrates key steps in our spatial standardisation procedure.

One other question is which figure should be used for the main manuscript figure. Perhaps the authors feel it would be misleading to select one of the 12 boxes and present that (if so ok leave things as they are, or perhaps even just switch Fig 2 and fig S3), but I think if it is made clear in the caption that all the other approaches show broadly the same result I think that is ok. The paper presents a convincing 'global' terrestrial diversity plot for the Phanerozoic, I know its not possible to actually do a global curve because of the nature of the data and analyses but at the moment I think having 12 virtually identical plots in the main results figures buries the message and impact of the result. From my perspective the ideal way to make very clear the main result would be to select one of the plots from supplementary figure 3 (maybe squares 3000 or sqs 3000) that has the model lines on it as the main results figure then put the rest in the supplement, or have the current fig 2 as a fig 3.

We have followed Reviewer 3's suggestion that we simplify the figure showing spatially-standardised regional-scale terrestrial tetrapod diversity. Our main-text figure (Fig. 4 in the revised MS) now shows only 2000 km MST distance (the same scale used for subclade patterns in Fig. 5), and additional spatial scales are shown in Fig. S9.

References

1. Peters, S. E. & Husson, J. M. 2017 Sediment cycling on continental and oceanic crust. *Geology* **45**, 323–326. (doi:10.1130/g38861.1)
2. Close, R. A. et al. 2019 Diversity dynamics of Phanerozoic terrestrial tetrapods at the local-community scale. *Nat. Ecol. Evol.* **3**, 590–597. (doi:10.1038/s41559-019-0811-8)
3. Rabosky, D. L. & Hurlbert, A. H. 2015 Species richness at continental scales is dominated by ecological limits. *Am. Nat.* **185**, 572–583. (doi:10.1086/680850)
4. Harmon, L. J. & Harrison, S. 2015 Species diversity is dynamic and unbounded at local and continental scales. *Am. Nat.* **185**, 584–593. (doi:10.1086/680859)

Appendix B

Dear Dr Hutchinson,

Thank you for conveying the good news that our work has been accepted for publication in *Proceedings of the Royal Society B*! We are very gratified that our revised manuscript met with the approval of the Associate Editor and reviewer.

We look forward to seeing our work in print! Thank you for overseeing the valuable peer-review process.

Sincerely, and on behalf of the authors,

Roger Close